# Beyond Student: An Asymmetric Network for Neural Network Inheritance

**Yiyun Zhou** ♠∗  **Jingwei Shi** ♡∗  **Mingjing Xu** ♣∗  **Zhonghua Jiang** ♠  **Jingyuan Chen** ♠†

♠Zhejiang University  ♡Shanghai University of Finance and Economics  ♣ Swansea University
{yiyunzhou, jiangzhonghua, jingyuanchen}@zju.edu.cn  shijingwei@stu.sufe.edu.cn
mingjing.xu@swansea.ac.uk

Demo Code: https://github.com/zyy-2001/InherNet-Demo

## Abstract

Knowledge Distillation (KD) has emerged as a powerful technique for model compression, enabling lightweight student networks to benefit from the performance of redundant teacher networks. However, the inherent capacity gap often limits the performance of student networks. Inspired by the expressiveness of pretrained teacher networks, a compelling research question arises: *is there a type of network that can not only inherit the teacher's structure but also maximize the inheritance of its knowledge?* Furthermore, *how does the performance of such an inheriting network compare to that of student networks, all benefiting from the same teacher network?* To further explore this question, we propose InherNet, a neural network inheritance method that performs asymmetric low-rank decomposition on the teacher's weights and reconstructs a lightweight yet expressive network without significant architectural disruption. By leveraging Singular Value Decomposition (SVD) for initialization to ensure the inheritance of principal knowledge, InherNet effectively balances depth, width, and compression efficiency. Experimental results across unimodal and multimodal tasks demonstrate that InherNet achieves higher performance compared to student networks of similar parameter sizes. Our findings reveal a promising direction for future research in efficient model compression beyond traditional distillation.

## 1 Introduction

In recent years, Knowledge Distillation (KD) (Hinton et al., 2015) has gained widespread attention and recognition in both academia and industry as an efficient model compression technique (Zhou et al., 2018; Xu et al., 2024b). Its core idea is to transfer knowledge from a large, pretrained teacher network to a lightweight student network, thereby improving the student's performance without significantly increasing additional computational cost. Initially, KD is mainly applied to unimodal tasks, such as visual tasks (Hinton et al., 2015; Jin et al., 2023) and textual tasks (Jiao et al., 2019; Kang et al., 2023). With the advancement of multimodal learning, KD has also been extended to multimodal scenarios for cross-modal knowledge transfer and fusion (Dai et al., 2022; Yang et al., 2024).

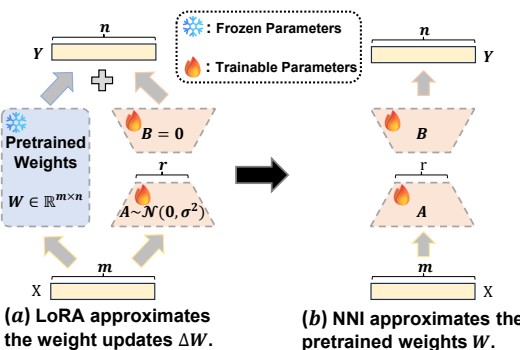

**(a)** LoRA approximates the weight updates $\Delta W$.

**(b)** NNI approximates the pretrained weights $W$.

Figure 1: The difference between LoRA and NNI.

---

∗Equal contribution.
†Corresponding author.

However, despite its effectiveness, a widely accepted view is that the performance of student networks is typically inferior to that of teacher networks due to limited capacity (Gou et al., 2021; Zhou et al., 2025b). This observation draws a parallel to Parameter-Efficient Fine-Tuning (PEFT) (Mangrulkar et al., 2022; Xu et al., 2023; Zhou et al., 2025f), especially the LoRA method (Hu et al., 2022). As illustrated in Figure 1 (a), given the pretrained weights $W \in \mathbb{R}^{m \times n}$, LoRA approximates the weight updates $\Delta W$ using a pair of low-rank matrices: $A \in \mathbb{R}^{m \times r}$ (initialized with Gaussian noise) and $B \in \mathbb{R}^{r \times n}$ (initialized with zeros), where $r \ll \min(m, n)$. This approach significantly reduces the number of trainable parameters from $m \times n$ to $r(m + n)$. However, LoRA generally still exhibits a performance gap compared to full fine-tuning (Zhang et al., 2023b; Ding et al., 2023; Han et al., 2024).

Given **the strong performance of teacher networks in KD, the difficulty of designing student networks to adapt to the teacher network (Gu & Tresp, 2020; Liu et al., 2022), and the parameter efficiency of low-rank approximations in LoRA**, we are naturally led to a more affinitive model compression approach: *Neural Network Inheritance* (**NNI**). Instead of training a separate student network, **NNI directly inherits the capacity of the teacher network in a low-rank form**. As shown in Figure 1 (b), NNI achieves model compression by performing low-rank decomposition on each layer of the teacher network and replacing the original weights with their decomposed counterparts. Crucially, unlike **LoRA which approximates weight updates $\Delta W$, InherNet approximates the full pre-trained weights** $W$, preserving the teacher model's expressiveness more directly.

Upon reviewing prior work, we find that a general and widely applicable method for NNI is lacking. **Existing related efforts are rather fragmented**, mainly focusing on early Convolutional Neural Network (CNN) architectures (Jaderberg et al., 2014; Zhang et al., 2015; Xu et al., 2019; Yang et al., 2020), and are mostly confined to vision tasks, with limited applicability to other modalities—particularly the increasingly prominent multimodal tasks. Moreover, **these methods (Lebedev et al., 2014; Hu et al., 2024) often rely on traditional tensor decomposition techniques** (*e.g.*, CP decomposition and MPO decomposition), which break a single layer into multiple sub-layers for compression, **often making networks deeper and narrower—raising risks of vanishing or exploding gradients and compromising stability and performance**. More importantly, there remains **a lack of investigation into the teacher inheritance under KD**, especially in terms of systematically comparing inherited and student networks in performance and efficiency.

To this end, we propose InherNet—a novel and general method for efficiently inheriting the knowledge and structure of teacher networks in KD. For knowledge inheritance, unlike prior decomposition-based methods (Jaderberg et al., 2014), InherNet leverages Singular Value Decomposition (SVD) for initialization (**SVD offline single-pass computation makes the training time negligible**), allowing the decomposition of only two layers without excessively deepening the network. For structure inheritance, InherNet is **architecture-agnostic** and applies low-rank approximation to convolutional or linear layers. Previous studies have shown that Mixture of Experts (MoE) models can significantly enhance performance across various tasks (Zhou et al., 2022; Tian et al., 2024; Gao et al., 2024; Cai et al., 2024). Inspired by this, InherNet adopts a similar design, enabling both deepening and widening of the network to mitigate the potential risks of gradient vanishing or explosion. This design enables InherNet to remain structurally aligned with the teacher network while also inheriting its representational capacity more faithfully. Notably, our experiments reveal a compelling insight: **InherNet converges significantly faster than traditional student networks**. We attribute this phenomenon to the extended SVD-based initialization and provide a detailed theoretical analysis to support this claim. Additionally, each layer of the teacher network inherited by InherNet uses a **fixed** asymmetric design (*i.e.*, a single dimensionality reduction matrix and multiple dimensionality expansion matrices). **We provide empirical evidence and theoretical support for this fixed design in Appendix A.**

To summarize, our contributions are in four aspects:

- We propose InherNet, a novel method that explicitly inherits both the knowledge and structure of powerful teacher networks in KD. Unlike implicit knowledge distillation, InherNet constructs a wider, deeper, yet lightweight network without the challenges of heterogeneous knowledge transfer.
- We uncover and analyze three pivotal insights regarding InherNet, as presented in Sec. 3.4. These findings illuminate the underlying principles of neural network inheritance and offer clear directions for future research in this area.

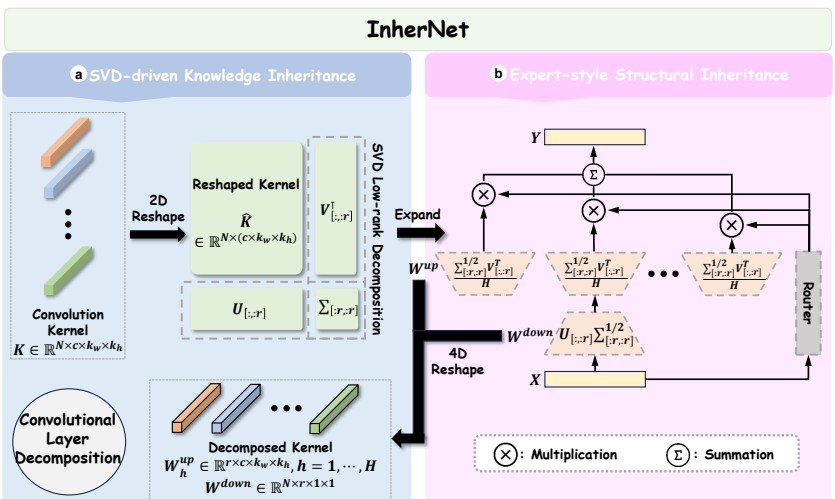

Figure 2: Overview of the proposed InherNet (*e.g.*, decomposition of a convolutional layer). Inher-Net consists of two parts: (a) Knowledge Inheritance and (b) Structure Inheritance. Note that the low-rank decomposition of linear layer is similar, with the key being to satisfy the properties of the 2D SVD operation.

- We provide extensive and rigorous theoretical analysis to explain the empirically observed phenomenon of InherNet's accelerated convergence with comparable parameter sizes. Our proofs formally attribute this efficiency advantage to its SVD-based initialization, establishing a solid theoretical foundation for its design.

- We conduct systematic experiments across various network architectures and modal tasks, including both unimodal and multimodal scenarios. The results consistently demonstrate that InherNet outperforms traditional student networks in both efficiency and effectiveness.

## 2 METHODOLOGY

In this section, we introduce our proposed InherNet method, which consists of knowledge inheritance and structure inheritance, as shown in Figure 2. Next, we will provide theoretical analyses of the method's convergence guarantees and parameter efficiency.

### 2.1 INHERNET

#### 2.1.1 KNOWLEDGE INHERITANCE

For a linear layer, consider a pretrained teacher network with a weight matrix $W \in \mathbb{R}^{m \times n}$. InherNet employs SVD to factorize $W$ into a compact low-rank approximation. Specifically, we have:

$$W = U\Sigma V^\top \approx U_r \Sigma_r V_r^\top, \tag{1}$$

where $U_r \in \mathbb{R}^{m \times r}$, $\Sigma_r \in \mathbb{R}^{r \times r}$, and $V_r \in \mathbb{R}^{n \times r}$ represent truncated matrices derived from the top-$r$ singular values and vectors. Therefore, the original layer is approximated by two consecutive low-rank layers, $U_r \Sigma_r$ and $\Sigma_r V_r^\top$. The approximation is supported by Theorem 2.1.

**Theorem 2.1** (Eckart-Young-Mirsky theorem). *Given a weight matrix $W \in \mathbb{R}^{m \times n}$ with singular value decomposition $W = U\Sigma V^\top$, the optimal rank-$r$ approximation in terms of Frobenius norm is given by: $W_r = U_{[:,:r]} \Sigma_{[:r,:r]} V_{[:,:r]}^\top$. Moreover, the approximation error is minimized, specifically:*

$$\|W - W_r\|_F = \sqrt{\sum_{i=r+1}^{\min(m,n)} \sigma_i^2(W)}, \tag{2}$$

*where $\sigma_i(W)$ denotes the $i$-th singular value of $W$ (Golub et al., 1987; Eckart & Young, 1936; Golub & Van Loan, 2013).*

For a convolutional layer, as shown in Figure 2 (a), formally, there exists a convolution kernel $K \in \mathbb{R}^{N \times c \times k_w \times k_h}$, where $n, c, k_w, k_h$ represent the number of kernels, the number of input channels, and the width and height of the filter, respectively. Our work focuses on the channel decomposition method (Zhang et al., 2015) to decompose the convolution layer. Specifically, $K$ is first reshaped into a 2D matrix $\hat{K} \in \mathbb{R}^{N \times (c \times k_w \times k_h)}$, and SVD initialization is applied. The resulting decomposed layers are reshaped to have convolutional kernel weights $W^{down} \in \mathbb{R}^{N \times r \times 1 \times 1}$ and $W^{up} \in \mathbb{R}^{r \times c \times k_w \times k_h}$.

### 2.1.2 STRUCTURE INHERITANCE

Unlike previous low-rank decomposition methods (Jaderberg et al., 2014; Zhang et al., 2015; Zhou et al., 2025c), we adopt an expert-style inheritance structure inspired by the Mixture of Experts (MoE) paradigm, which has been widely proven effective across various domains (Tian et al., 2024; Gao et al., 2024; Cai et al., 2024). As illustrated in Figure 2 (b), in order to strike a good balance between performance and efficiency, we design an asymmetric expert-head structure. Specifically, given input features $X \in \mathbb{R}^m$, the output $Y \in \mathbb{R}^n$ of InherNet is formulated as:

$$Y = \sum_{h=1}^{H} G_h(X) \cdot W_h^{up} \left( W^{down}(X) \right), \tag{3}$$

where $H$ denotes the number of expert heads, $W^{down}$ and $W_h^{up}$ represent the "down" projection of the backbone branch and the "up" projection of the $h$-th expert head, respectively. $W^{down}$ and $W_h^{up}$ are initialized using $U_{[:,:r]}\Sigma_{[:r,:r]}^{1/2}$ and $\frac{\Sigma_{[:r,:r]}^{1/2} V_{[:,:r]}^{\top}}{H}$ from Eq. 1, and the matrix slicing notations follow the same convention as in PyTorch. $G(X)$ represents adaptive gating fusion weights computed as:

$$G(X) = \mathrm{softmax}\left(W^g(X)\right), \tag{4}$$

where $W^g \in \mathbb{R}^{m \times H}$ is a learnable parameter.

The InherNet's procedure can be found in Appendix B.1. We also provide empirical and theoretical support for InherNet's fixed asymmetric structural design in Appendix A. Note that InherNet inherently benefits from SVD and gating mechanism, theoretically enabling excellent convergence and parameter efficiency, as rigorously analyzed in Sec. 2.2 and Sec. 2.3. By directly inheriting structural knowledge through decomposition and adaptively aggregating expert heads, InherNet provides an efficient and expressive solution, as demonstrated by our experiments in Sec. 3.

### 2.2 THEORETICAL ANALYSIS OF CONVERGENCE GUARANTEES

We analyze the convergence properties of InherNet, focusing on its structured low-rank decomposition, multi-head gating mechanism, and SVD-based initialization. Our goal is to establish that, despite the compression from low-rank projections, InherNet has stable and efficient optimization under stochastic gradient descent (SGD).

Let $\mathcal{L}(\theta) = \mathbb{E}_{(x,y) \sim \mathcal{P}}[\ell(f_\theta(x), y)]$ be the expected loss, where $(x, y) \sim \mathcal{P}$ denotes data sampled from the underlying distribution. Let $\theta \in \mathbb{R}^d$ denote the parameters of an InherNet, and let $\theta^{(t)}$ be the model parameters at iteration $t$ updated via stochastic gradients.

We impose the following assumptions, which are common in the analysis of stochastic optimization for nonconvex problems (Patel et al., 2022; Xu et al., 2024a; Lei et al., 2019; Pham et al., 2020):

**Assumption 2.1** (Lipschitz Continuity). The expected loss gradient is $L$-Lipschitz continuous:

$$\|\nabla \mathcal{L}(\theta) - \nabla \mathcal{L}(\theta')\| \leq L\|\theta - \theta'\|, \quad \forall \theta, \theta' \in \mathbb{R}^d. \tag{5}$$

**Assumption 2.2** (Bounded Variance). The stochastic gradient $\nabla_\theta \ell(f_\theta(x), y)$ is unbiased and has bounded variance:

$$\mathbb{E}[\nabla_\theta \ell(f_\theta(x), y)] = \nabla \mathcal{L}(\theta), \quad \mathbb{E}\left[\|\nabla_\theta \ell - \nabla \mathcal{L}(\theta)\|^2\right] \leq \sigma^2. \tag{6}$$

**Assumption 2.3** (Bounded Representations). There exist constants $B_x, B_f > 0$ such that $\|x\| \leq B_x$ and $\|f_\theta(x)\| \leq B_f$ for all $(x, y) \sim \mathcal{P}$.

### 2.2.1 GRADIENT DECOMPOSITION UNDER INHERNET

InherNet models each module as a low-rank gated composition:

$$f_\theta(x) = \sum_{h=1}^{H} G_h(x; \theta_g) \, W_h^{\text{up}}(W^{\text{down}}(x)), \tag{7}$$

where $W^{\text{down}}$ is the shared projection, $W_h^{\text{up}}$ is the $h$-th expert head, and $G_h(.)$ is a gating weight.

**Lemma 2.2** (Gradient Decomposition). *Under the InherNet parameterization, the gradient of the loss splits naturally into contributions from each head and from the gating network. The total gradient $\nabla_\theta \ell(f_\theta(x), y)$ decomposes into:*

$$\nabla_\theta \ell\big(f_\theta(x), y\big) = \sum_{h=1}^{H} G_h\big(x; \theta_g\big) \, \nabla_{\theta_h} \ell\big(f_\theta(x), y\big) + \sum_{h=1}^{H} \Delta_h \, \nabla_{\theta_g} G_h\big(x; \theta_g\big), \tag{8}$$

*where $\theta_h$ and $\theta_g$ are the parameters of expert head $h$ and the gating network, respectively, and $\Delta_h = \frac{\partial \ell(f_\theta(x), y)}{\partial f_h(x)}$ represents the gradient of the loss with respect to the output of the $h$-th head, with $f_h(x) = W_h^{up}(W^{down}(x))$.*

This gradient routing allows InherNet to specialize experts during training, directing gradient flow to the most relevant parameters for each input. It is beneficial when compressing teacher networks.

### 2.2.2 EFFECT OF SVD-BASED INITIALIZATION AND CONVERGENCE GUARANTEE

Let $W$ denote the pretrained weight from a teacher network. InherNet initializes each module using truncated SVD. The initialized parameters are: $W^{\text{down}} = U_r \Sigma_r^{1/2}, \quad W_h^{\text{up}} = \frac{1}{H} \Sigma_r^{1/2} V_r^\top$.

**Proposition 2.3** (Stability via Orthonormal Initialization). *If $U_r$ and $V_r$ are initialized with orthonormal columns, then under Assumption 2.1, the effective gradient Lipschitz constant is reduced from $L$ to $L' \approx L/\kappa$, where $\kappa$ is the condition number of $W$.*

This improved conditioning reduces curvature imbalance during early training and stabilizes convergence, particularly when the teacher model is well-trained (Wang et al., 2024; Lim & Ye, 2024; Saxe et al., 2013).

**Theorem 2.4** (Non-Convex Convergence). *Under Assumptions 2.1–2.3, and adopting a diminishing learning rate schedule $\eta_t = \frac{\eta}{\sqrt{t}}$, the sequence of parameters $\{\theta^{(t)}\}$ generated by InherNet satisfies the following convergence rate:*

$$\frac{1}{T} \sum_{t=1}^{T} \mathbb{E}\left[\|\nabla\mathcal{L}(\theta^{(t)})\|^2\right] = \mathcal{O}\left(\frac{1}{\sqrt{T}}\right). \tag{9}$$

The convergence rate matches classical result for non-convex optimization problems (Xu et al., 2024a). Because of the SVD-based orthonormal initialization and adaptive gating mechanism, InherNet achieves convergence rate of $\mathcal{O}(1/\sqrt{T})$ with practical scaling as $\mathcal{O}(L/\kappa)$. The improved convergence behavior stems from Proposition 2.3, where the Lipschitz constant for gradient updates is reduced from $L$ to $L' \approx L/\kappa$. This reduction in curvature imbalance commits to lower gradient variance and faster gradient descent in early stages, as empirically verified in Sec. 3.4.

### 2.3 THEORETICAL PROOF OF PARAMETER EFFICIENCY

We analyze parameter efficiency of InherNet, establishing theoretical guarantees on both parameter reduction and representational power preservation. Our analysis formalizes the efficiency-expressivity tradeoff through three metrics:

**Definition 2.1** (Parameter Efficiency (PE)). For network $f_\theta$ parameterized by $\theta \in \mathbb{R}^p$, we define $\text{PE}(f_\theta) = \frac{\text{Representational Capacity}(f_\theta)}{|\theta|}$. Representational capacity measures the network's approximation ability and $|\theta|$ denotes parameter count.

**Definition 2.2** (Approximation Error). Given a target function $f^*$ and a neural network $f_\theta$, the approximation error is $\mathcal{E}(f_\theta, f^*) = \mathbb{E}_{x \sim \mathcal{D}}[\|f_\theta(x) - f^*(x)\|^2]$, where $\mathcal{D}$ is the data distribution.

**Definition 2.3** (Expressivity-to-Parameter Ratio (EPR)). The EPR of network $f_\theta$ with respect to function class $\mathcal{F}$ is $\text{EPR}(f_\theta, \mathcal{F}) = \frac{1}{|\theta| \inf_{f^* \in \mathcal{F}} \mathcal{E}(f_\theta, f^*)}$, measuring the inverse of minimum achievable approximation error per parameter.

### 2.3.1 COMPRESSION BOUNDS FOR INHERNET

We first quantify the parameter reduction achieved by InherNet relative to a standard network, which directly improves the PE as defined in Definition 2.1.

**Theorem 2.5** (Parameter Reduction Bounds). *Consider a layer $W \in \mathbb{R}^{m \times n}$. The InherNet parameterization with rank $r$ and $H$ heads uses $H\,r\,(m+n) + H\,(r+1)$ trainable parameters, yielding a compression ratio:*

$$\rho = \frac{m\,n}{H\,r\,(m+n) + H\,(r+1)} = \frac{m\,n}{H\,r\,(m+n)}\left(1 + O\left(\tfrac{1}{m+n}\right)\right) \tag{10}$$

*Thus, it achieves an approximately $\rho$-fold increase in Definition 2.1, under the assumption that a rank-$H\,r$ subspace preserves the layer's representational capacity.*

To ensure low Approximation Error (Definition 2.2) despite compression, we establish:

**Lemma 2.6** (Spectral Energy Preservation). *Let $W \in \mathbb{R}^{m \times n}$ have singular values $\sigma_1 \geq \sigma_2 \geq \cdots \geq \sigma_{\min(m,n)}$. If the rank $r$ is selected such that:*

$$\frac{\sum_{i=1}^{r} \sigma_i^2}{\sum_{i=1}^{\min(m,n)} \sigma_i^2} \geq 1 - \epsilon \tag{11}$$

*then by Theorem 2.1, the Frobenius-norm error of the optimal rank-$r$ approximation $W_r$ is bounded by: $\|W - W_r\|_F^2 \leq \epsilon \|W\|_F^2$. Consequently, the Definition 2.2 between an InherNet-compressed layer and the original layer is bounded by $\epsilon$ under the $L_2$ norm squared.*

**Proposition 2.7** (Knowledge Preservation Rate). *Let $f_W$ be a teacher network with weight matrices $\{W_l\}$ and $f_r$ be the corresponding InherNet with rank $r$. The functional similarity between $f_W$ and $f_r$ is lower-bounded by:*

$$Sim(f_W, f_r) \geq 1 - \sum_{l=1}^{L} \alpha_l \left(1 - \frac{\sum_{i=1}^{r} \sigma_{l,i}^2}{\sum_{i=1}^{\min(m_l, n_l)} \sigma_{l,i}^2}\right) \tag{12}$$

*where $\alpha_l$ represents the layer's influence on output, and $\sigma_{l,i}$ are the singular values of $W_l$. This bound is monotonically increasing with $r$ but not with $H$, establishing that rank is the primary metric of inherited knowledge preservation.*

### 2.3.2 REPRESENTATIONAL POWER PRESERVATION

InherNet also maintains the representational power of the original network, preserving a beneficial EPR (Definition 2.3):

**Theorem 2.8** (Representational Power Preservation). *Let $\mathcal{F}_W$ be functions representable by a standard network with weights $\{W_l\}_{l=1}^{L}$. For any $f^* \in \mathcal{F}_W$ and $\delta > 0$, there exists an InherNet network with appropriate ranks $\{r_l\}$ and head counts $\{H_l\}$ such that the Approximation Error (Definition 2.2) satisfies:*

$$\mathcal{E}(f_{InherNet}, f^*) \leq \delta \tag{13}$$

*while the parameter count is reduced by $\Omega(\min_l \frac{\min(m_l, n_l)}{r_l H_l})$ relative to the standard network. Consequently, the InherNet architecture attains an EPR (Definition 2.3) higher than that of the standard architecture by at least the same factor.*

This preservation stems from two complementary mechanisms: 1. **Layer-wise low-rank approximation:** Capturing the dominant spectral components of each weight matrix; 2. **Multi-head specialization:** Allowing different heads to specialize on different input subspaces.

**Corollary 2.9** (Universal Approximation). *InherNet retains the universal function approximation property of standard neural networks while using asymptotically fewer parameters. For any continuous target function, a sufficiently large InherNet network can approximate it arbitrarily well (to Approximation Error $\delta \to 0$) with significantly higher Parameter Efficiency and EPR than a standard network of comparable accuracy.*

Table 1: Comparison of top-1 Accuracy (%) on the CIFAR-100 validation set between InherNet and previous state-of-the-art knowledge distillation methods. **Best results** are in bold, and the second-best are underlined.

| KD Type | | ResNet32×4 | VGG13 | WRN-40-2 | WRN-40-2 | ResNet56 | ResNet110 | ResNet110 |
|---|---|---|---|---|---|---|---|---|
| KD Type | Teacher | ResNet32×4 79.42 | VGG13 74.64 | WRN-40-2 75.61 | WRN-40-2 75.61 | ResNet56 72.34 | ResNet110 74.31 | ResNet110 74.31 |
| | Student | ResNet8×4 72.50 | VGG8 70.36 | WRN-40-1 71.98 | WRN-16-2 73.26 | ResNet20 69.06 | ResNet32 71.14 | ResNet20 69.06 |
| Feature | CRD | 75.51 | 73.94 | 74.14 | 75.48 | 71.16 | 73.48 | 71.46 |
| | OFD | 74.95 | 73.95 | 74.33 | 75.24 | 70.98 | 73.23 | 71.29 |
| | ReviewKD | 75.63 | 74.84 | 75.09 | 76.12 | 71.89 | 73.89 | 71.34 |
| | SimKD | 78.08 | 74.89 | 74.53 | 75.53 | 71.05 | 73.92 | 71.06 |
| | CAT-KD | 76.91 | 74.65 | 74.82 | 75.60 | 71.62 | 73.62 | 71.37 |
| Logit | KD | 73.33 | 72.98 | 73.54 | 74.92 | 70.66 | 73.08 | 70.67 |
| | KD+CTKD | 73.39 | 73.52 | 73.93 | 75.45 | 71.19 | 73.52 | 70.99 |
| | DKD | 76.32 | 74.68 | 74.81 | 76.24 | 71.97 | 74.11 | 71.06 |
| | MLKD | 77.08 | 75.18 | 75.35 | 76.63 | 72.19 | 74.11 | 71.89 |
| | MLKD+Logit Std. | 78.28 | 75.22 | 75.56 | **76.95** | 72.33 | 74.32 | 72.27 |
| InherNet | Small | 77.57 | **75.68** | 76.04 | 76.04 | 72.67 | 74.13 | 74.13 |
| | Large | **78.53** | 75.16 | **76.39** | 76.39 | **73.67** | **75.88** | **75.88** |

**Proposition 2.10** (Diminishing Returns of Multiple Heads). *For a given layer with rank $r$, the marginal reduction in approximation error from adding the $(H+1)$-th head satisfies:*

$$\mathcal{E}(r, H) - \mathcal{E}(r, H+1) \leq \frac{c}{H^2}\mathcal{E}(r, 1) \tag{14}$$

*where $c$ is a constant depending on input distribution characteristics, and $\mathcal{E}(r, H)$ denotes the approximation error with rank $r$ and $H$ heads. This establishes that benefits from additional heads diminish at a rate of $\Omega(1/H^2)$.*

The theoretical results explain empirical insights in Sec. 3.4: (1) rank plays the primary role in preserving teacher knowledge as formalized in Proposition 2.7, and (2) while multiple heads outperform a single head, adding more heads yields diminishing returns as quantified in Proposition 2.10. **Detailed theoretical proofs and extended discussions are provided in Appendix B**.

## 3 EXPERIMENTS

We provide all implementation details (including a detailed introduction to the datasets and baselines) of the experiments, along with the parameter sizes of the teacher and student networks and InherNet (introducing two variants, InherNet $_{\text{Small}}$ and InherNet $_{\text{Large}}$), in Appendix C.1.

### 3.1 VISION: IMAGE CLASSIFICATION

In this task, we conduct extensive experiments to validate the advantages of the proposed InherNet over traditional KD methods. The experimental setup is as follows:

**Datasets.** We use two widely adopted image classification datasets—CIFAR-100 (Krizhevsky & Hinton, 2009) and ImageNet (Russakovsky et al., 2015).

**Baselines.** We compare against several mainstream KD methods, including logit-based methods: KD (Hinton et al., 2015), CTKD (Li et al., 2023b), DKD (Zhao et al., 2022), MLKD (Jin et al., 2023), and Logit Std. (Sun et al., 2024); as well as feature-based methods: CRD (Peng et al., 2019), OFD (Heo et al., 2019), ReviewKD (Chen et al., 2021), SimKD (Chen et al., 2022), and CAT-KD (Guo et al., 2023).

### 3.1.1 RESULTS ON CIFAR-100

Table 1 compares the top-1 accuracy of InherNet with previous state-of-the-art KD methods. All methods used 240 epochs, except for MLKD for 480 epochs. The choice of 240 epochs is because all methods, including InherNet (even with faster convergence due to SVD initialization, as validated by Insight 3 in Appendix C.4 and Figure 5), converge within this period, except for MLKD. With 240 epochs, MLKD has not yet converged; using ResNet56 as the teacher and ResNet20 as the student, MLKD even achieves lower accuracy than the student's. This result is consistent with previous

studies Jin et al. (2023); Sun et al. (2024). In contrast to such unfair evaluations, we exclude the unfair baselines in our other analyses. The main findings are as follows:

- Among all baselines, the recently proposed MLKD and Logit Std. achieve the best performance. This may be due to the unfair comparison settings in previous works (Jin et al., 2023; Sun et al., 2024), where these methods are trained for more epochs, as they converge more slowly and perform poorly with fewer epochs (*e.g.*, with 240 epochs, MLKD only achieves 60.05% accuracy when using ResNet56 as teacher and ResNet20 as student). In contrast, other KD methods use fewer epochs but show limited improvement.

- Under fair experimental settings (excluding MLKD and Logit Std.), a significant performance gap remains between student and teacher networks. This is not only due to the limited capacity and architecture of the student network, but also because of the underlying objective of KD—using the teacher network as a reference to guide the student network toward its performance limit.

- The proposed InherNet strikes a balance between efficiency and effectiveness. With fewer epochs, the lightweight InherNet $_{\text{Small}}$ achieves competitive performance compared to the state-of-the-art KD method Logit Std., while the slightly larger InherNet $_{\text{Large}}$ consistently outperforms other methods and significantly narrows the performance gap with the teacher network. We attribute this to the proposed extended SVD initialization, which inherits most of the teacher's forward knowledge, and the asymmetric MoE inheritance structure, which enables robust generalization during training. Moreover, **InherNet empirically shows faster convergence** (Sec. 3.4) and **better parameter efficiency**, backed by theoretical convergence guarantees (Sec. 2.2) and parameter efficiency proofs (Sec. 2.3).

### 3.1.2 RESULTS ON IMAGENET

In Apppendix C.2, we provide experimental results on ImageNet, showing that InherNet $_{\text{Large}}$ continues to demonstrate leading performance on the large-scale dataset, while InherNet $_{\text{Small}}$ achieves a good balance between efficiency and effectiveness. All methods are trained for 100 epochs.

### 3.2 LANGUAGE: GLUE BENCHMARK

In this task, we evaluate InherNet's performance on multiple text tasks. The experimental setup is as follows:

**Datasets.** To remain consistent with prior research (Raffel et al., 2020), we train and evaluate InherNet's performance on the GLUE benchmark (Wang et al., 2019).

**Baselines.** Our experiments are based on the T5 model (Raffel et al., 2020) implemented in HuggingFace (Wolf et al., 2020). Due to the lack of widely known and reproducible T5-based KD methods, we use the traditional KD method (Hinton et al., 2015), with a teacher-student setup of T5-Base/T5-Small with parameters 222M/60M.

Beyond the T5 experiments, we also apply InherNet to a larger decoder-only LM, LLaMA-2-7B Raffel et al. (2020); Touvron et al. (2023). For such autoregressive LMs, conventional token-level KD is difficult to deploy due to the 32K-dimensional vocabulary softmax, strict teacher-forcing requirements, and tokenizer mismatches between teacher and student, so we do not include standard KD baselines here. Instead, we compare a self-distilled LLaMA-2-7B baseline (teacher and student sharing the same weights, offering no compression) with an InherNet model constructed from the same teacher. As detailed in Appendix C.3, the resulting $4.2$B-parameter model slightly outperforms the 7B teacher on GSM8K while maintaining competitive performance on MMLU-style evaluations, indicating that our inheritance mechanism scales to large Transformer LMs without relying on KD.

Table 2 shows the performance of InherNet and the distillation method on GLUE tasks. As shown in the table, there is still a performance gap between the distilled student network and the teacher network. However, the InherNet method we proposed narrows this gap through explicit inheritance, which distinguishes it from the traditional implicit learning approach used in KD.

Table 2: The performance (%) of InherNet and the distillation method on GLUE tasks. For MRPC, QQP, SST-2, MNLI, RTE, and QNLI, we report Accuracy. For CoLA, we report the Matthews correlation coefficient. For STS-B, we report Pearson and Spearman correlation coefficients.

| Method | MRPC | QQP | SST-2 | MNLI | RTE | QNLI | COLA | STS-B |
|---|---|---|---|---|---|---|---|---|
| T5-Base | 89.21 | 86.54 | 90.71 | 54.93 | 77.98 | 90.13 | 57.00 | 89.04 / 88.88 |
| T5-Small | 88.38 | 81.05 | 87.78 | 53.67 | 72.71 | 87.97 | 48.12 | 85.55 / 86.50 |
| T5-Small+KD | 88.23 | **84.92** | 89.01 | 53.81 | **74.31** | 88.76 | 49.31 | **88.05 / 87.85** |
| InherNet | **88.74** | 84.69 | **89.07** | **56.34** | 72.93 | **90.29** | 50.63 | 86.70 / 86.36 |

Table 3: Comparison of cross-modal retrieval on the CC3M validation set and zero-shot classification on ImageNet validation set performance (%) between InherNet and the student networks distilled with CLIP-KD trained on CC3M dataset.

| Method | I2T | | | T2I | | | Classification | |
|---|---|---|---|---|---|---|---|---|
| | R@1 | R@5 | R@10 | R@1 | R@5 | R@10 | top-1 | top-5 |
| ResNet-101 | 30.58 | 53.95 | 63.49 | 29.31 | 54.13 | 63.31 | 15.70 | 32.75 |
| ResNet-18 | 31.03 | 56.41 | 65.82 | 30.78 | 55.74 | 65.35 | 15.93 | 33.83 |
| EfficientNet-B0 | **33.65** | **58.72** | **68.28** | 32.65 | 58.21 | 67.92 | 17.30 | 35.75 |
| MobileNetV3 | 29.07 | 53.43 | 63.59 | 28.10 | 53.11 | 63.35 | 14.59 | 31.68 |
| InherNet | 32.17 | 57.17 | 67.82 | **33.01** | **59.88** | **68.72** | **17.39** | **36.65** |

## 3.3 VISION ⇔ LANGUAGE: MULTI-MODAL EVALUATION

In this task, we validate the effectiveness of InherNet in multimodal scenarios and compare it with the recent CLIP-KD (Yang et al., 2024). The experimental setup is as follows:

**Dataset.** We use Conceptual Captions 3M (CC3M) (Sharma et al., 2018) for visual-language pre-training. Additionally, we use the ImageNet validation set for zero-shot classification, with the evaluation protocol consistent with CLIP-related works (Wei et al., 2022; Li et al., 2023a).

**Baselines.** In this task, ResNet-101 serves as the teacher network, while ResNet-18 (He et al., 2016), EfficientNet-B0 (Tan & Le, 2019) and MobileNetV3 (Howard et al., 2019) are used as students.

Table 3 shows the cross-modal retrieval performance on the CC3M validation set and zero-shot classification performance on ImageNet validation set for InherNet and its student network distilled through CLIP-KD trained on the CC3M dataset. The proposed InherNet generally outperforms other baseline methods in various multimodal tasks, with its performance consistently significantly higher than that of the teacher network, further validating the effectiveness of InherNet in multimodal scenarios.

## 3.4 INSIGHTS ON INHERNET

Based on the image classification task, we conduct analytical experiments under the teacher-student setup of ResNet56/ResNet20. We provide experimental details and results for all insights in Appendix C.4. The observed insights are as follows.

**Insight 1.** *Distillation benefits small-scale InherNet, but harms the performance of large-scale InherNet.* It is aligned with our theory: as the rank $r$ grows, the InherNet $f_r$ converges in function space toward the teacher $f_W$ in Proposition 2.7. In the high-rank regime, the task loss alone can drive it to that generalize slightly better than the teacher. In contrast, the KD loss penalizes any deviation from the teacher's logits, acting as an overly strong regularizer. More details are shown in Appendix B.4.

**Insight 2.** *Rank significantly affects model performance; using multiple expert heads performs better than a single head.*

**Insight 3.** *Compared to traditional distillation methods, InherNet converges significantly faster.*

## 3.5 ABLATION STUDY

We construct three variants of InherNet $_{\text{Large}}$ with the teacher network ResNet56 on CIFAR-100 to investigate the critical roles of different components. Specifically, "w.o. svd" denotes the absence of SVD initialization, "w.o. gate"

Table 4: Ablation study of different variants on InherNet.

| Variants | ResNet32×4 | WRN-40-2 | ResNet56 | ResNet110 |
|---|---|---|---|---|
| w.o. svd | 76.68 | 75.42 | 69.35 | 74.13 |
| w.o. gate | 78.17 | 76.08 | 73.22 | 75.64 |
| w. sym. | 77.92 | 75.98 | 73.28 | 75.45 |
| InherNet | **78.53** | **76.39** | **73.67** | **75.88** |

removes the gating mechanism, and "w. sym." adopts a symmetric structure (with two branches and two expert heads, similar to LoRA+MoE (Liu et al., 2024)). The experimental results are shown in Table 4, from which we observe the following: (1) SVD initialization contributes the most to performance improvement. This is not only because it allows the network to inherit a substantial amount of beneficial knowledge from the teacher network, but also because it stabilizes training, accelerates convergence, and helps the model find the optimal solution more quickly and accurately, as illustrated in Insight 3 of Sec. 3.4. (2) The gating mechanism enables effective dynamic routing in most cases, and it is especially crucial for lightweight inherited networks. (3) Under the same parameter size, the symmetric architecture underperforms compared to asymmetric ones. This is because the asymmetric architecture introduces a structural inductive bias (Parton & Engelbrecht, 2020), which enhances the diversity among experts.

## 4 RELATED WORK

We summarize previous research of knowledge distillation (KD) (Hinton et al., 2015) and low-rank decomposition (Denton et al., 2014); extensive discussion and comprehensive comparisons appear in Appendix D.

## 5 CONCLUSION

This paper introduces InherNet, a novel approach beyond student that directly inherits both the structure and knowledge of teacher networks. By applying low-rank approximation and leveraging singular value decomposition, InherNet efficiently compresses models while maintaining the expressiveness and performance of the teacher network. Our method demonstrates faster convergence compared to traditional student networks and offers a more stable and effective solution for model compression.

## ACKNOWLEDGMENTS

This research was partially supported by grants from the "Pioneer" and "Leading Goose" R&D Program of Zhejiang under Grant No. 2025C02022, the National Natural Science Foundation of China (No.62307032), and Shanghai Rising-Star Program (23QA1409000). Additionally, this work was partially supported by the UK Engineering and Physical Sciences Research Council (EPSRC) [EP/W524694/1].

## 6 ETHICS STATEMENT

This work is guided by the ethical principles outlined in the ICLR Code of Ethics, prioritizing responsible research practices, scientific rigor, and the broader well-being of society. We recognize the global community of stakeholders in machine learning research and aim to ensure our contributions are beneficial to society while minimizing potential risks. Our research adheres to high standards of integrity, transparency, and reproducibility, with methods and results presented honestly and accurately. We have carefully considered the wider implications of our work, addressing potential concerns related to privacy, safety, and fairness, and have consulted with domain experts to mitigate any unintended consequences. All data used in this study has been handled in line with ethical approvals, safeguarding privacy and confidentiality. We are committed to promoting inclusivity, avoiding bias, and ensuring that our findings are both accessible and socially responsible.

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

## 7 Appendix

We present the following additional information to include and clarify details that could not be fully covered in the main text.

## A Fixed Asymmetric Design

### A.1 Empirical Evidence

We attempt to invert InherNet into InherNet$^{\circlearrowleft}$ (*i.e.*, multiple single-dimensionality reduction matrices followed by a dimensionality expansion matrix). We compare the performance of InherNet$^{\circlearrowleft}$ and InherNet across various tasks, as shown in Tables 5, 6, and 7. From the results, we observe that InherNet$^{\circlearrowleft}$ consistently underperforms compared to InherNet across all tasks. The performance gap is especially pronounced in text-related tasks, which may be attributed to the fact that, in such tasks, the linear layers adapted for inheritance more effectively preserve and replicate the knowledge from the teacher network. In contrast, convolutional neural networks, due to their higher dimensionality and the necessity of reshaping operations, may suffer greater information loss. We plan to explore further optimization strategies for inheriting convolutional neural networks as a direction for future work.

Table 5: Comparison of top-1 Accuracy (%) on the CIFAR-100 validation set between InherNet$^{\circlearrowleft}$ and InherNet.

| Method | | ResNet32×4 | VGG13 | WRN-40-2 | WRN-40-2 | ResNet56 | ResNet110 | ResNet110 |
|---|---|---|---|---|---|---|---|---|
| | Teacher | 79.42 | 74.64 | 75.61 | 75.61 | 72.34 | 74.31 | 74.31 |
| | | ResNet8×4 | VGG8 | WRN-40-1 | WRN-16-2 | ResNet20 | ResNet32 | ResNet20 |
| | Student | 72.50 | 70.36 | 71.98 | 73.26 | 69.06 | 71.14 | 69.06 |
| InherNet$^{\circlearrowleft}$ | Small | 75.19 | **75.84** | 74.17 | **76.53** | 71.39 | 71.68 | 71.68 |
| | Large | **78.79** | 71.26 | 75.24 | 76.17 | 73.38 | 74.95 | 74.95 |
| InherNet | Small | **77.57** | 75.68 | **76.04** | 76.04 | **72.67** | **74.13** | **74.13** |
| | Large | 78.53 | **75.16** | **76.39** | 76.39 | **73.67** | **75.88** | **75.88** |

Table 6: The performance (%) of InherNet$^\circlearrowleft$ and InherNet on GLUE tasks.

| Method | MRPC | QQP | SST-2 | MNLI | RTE | QNLI | COLA | STS-B |
|---|---|---|---|---|---|---|---|---|
| T5-Base | 89.21 | 86.54 | 90.71 | 54.93 | 77.98 | 90.13 | 57.00 | 89.04 / 88.88 |
| InherNet$^\circlearrowleft$ | 87.36 | 83.50 | 87.69 | 56.27 | 72.86 | 89.74 | 50.07 | 86.55 / 85.73 |
| InherNet | **88.74** | **84.69** | **89.07** | **56.34** | 72.93 | **90.29** | 50.63 | **86.70 / 86.36** |

Table 7: Comparison of cross-modal retrieval on the CC3M validation set and zero-shot classification on ImageNet validation set performance (%) between InherNet$^\circlearrowleft$ and InherNet.

| Method | I2T | | | T2I | | | Classification | |
|---|---|---|---|---|---|---|---|---|
| | R@1 | R@5 | R@10 | R@1 | R@5 | R@10 | top-1 | top-5 |
| ResNet-101 | 30.58 | 53.95 | 63.49 | 29.31 | 54.13 | 63.31 | 15.70 | 32.75 |
| InherNet$^\circlearrowleft$ | 28.76 | **57.46** | 64.17 | 30.39 | 59.25 | 65.81 | 17.24 | 36.02 |
| InherNet | **32.17** | 57.17 | **67.82** | **33.01** | **59.88** | **68.72** | **17.39** | **36.65** |

## A.2 THEORETICAL SUPPORT

A pretrained teacher network's weight $W \in \mathbb{R}^{m \times n}$ can be optimally approximated as $W \approx U_r \Sigma_r V_r^\top$, derived from the SVD, where $r \ll \min(m, n)$. A critical design choice arises when integrating this low-rank factorization with MoE principles to handle heterogeneous data distributions. We analyze two possible asymmetric architectures:

- **One-Down-Many-Ups**: A single, shared dimension-reducing projection $W^{\text{down}}$ feeds into multiple specialized dimension-increasing expert heads $\{W_h^{\text{up}}\}_{h=1}^H$, as shown in Figure 2.

- **Many-Downs-One-Up**: Multiple specialized projections $\{W_h^{\text{down}}\}_{h=1}^H$ are aggregated and passed through a single shared reconstruction layer $W^{\text{up}}$.

We found that the first configuration is theoretically superior, and we ground our analysis in the Information Bottleneck (IB) principle (Koch & Ullman, 1987; Alemi et al., 2016; Li et al.).

The IB principle frames learning as a problem of compression. Given an input $X$ and a target output $Y$, the goal is to learn a compressed representation, or bottleneck, $Z$, that is maximally informative about $Y$ while being minimally informative about $X$. This trade-off is formalized by minimizing the Lagrangian:

$$\mathcal{L}_{\text{IB}}(p(z|x)) = I(X; Z) - \beta I(Z; Y) \tag{15}$$

where $I(\cdot; \cdot)$ denotes mutual information and $\beta$ is a Lagrange multiplier controlling the trade-off between compression ($I(X; Z)$) and prediction ($I(Z; Y)$). The optimal representation $Z$ is thus a minimal sufficient statistic of $X$ for predicting $Y$.

### A.2.1 INFORMATION-THEORETIC ANALYSIS OF ASYMMETRIC ARCHITECTURES

We now apply the IB framework to analyze the two competing architectures. Let the input features be $X$ and the final output be $Y$.

**The One-Down-Many-Ups Configuration as a Unified Bottleneck** This architecture, as defined in InherNet, is given by:

$$Y = \sum_{h=1}^H G_h(X) \cdot W_h^{\text{up}} \left( W^{\text{down}}(X) \right) \tag{16}$$

Let us define the bottleneck variable as the output of the shared projection: $Z \triangleq W^{\text{down}}(X)$. The IB objective for this architecture is to learn the parameters of $W^{\text{down}}$ (which define the mapping $p(z|x)$) that minimize $\mathcal{L}_{\text{IB}}$ from Eq. 15.

**Theoretical Analysis.** The one-down-many-ups architecture imposes a single shared bottleneck: all information from $X$ to $Y$ must pass through the same low-dimensional $Z$. This means $Z$ must serve as the representation for the *entire mixture* of $H$ experts. Consequently, to maximize the predictive information $I(Z; Y)$, $Z$ is forced to encode only those features of $X$ that are salient for

predicting $Y$ across all expert domains, while filtering out any spurious or task-specific details (Ye et al., 2024; Zhou et al., 2025e;d; Lv et al., 2025; Zhou et al., 2025a). In effect, the model learns to focus on the joint task distribution $p(y|x) = \sum_{h=1}^{H} p(h|x)\, p(y|x, h)$, where $p(h|x)$ is given by the gating function $G_h(X)$. Since the architecture essentially forms a Markov chain $X \rightarrow Z \rightarrow Y$ (Norris, 1998), the Data Processing Inequality guarantees that $I(Z; Y) \leq I(X; Y)$. The IB objective drives $Z$ to approach this upper bound on predictive power while minimizing $I(X; Z)$, yielding a minimal sufficient representation of $X$.

This unified bottleneck encourages $Z$ to become a robust, canonical representation of shared knowledge that all experts can rely on. Each expert head $W_h^{\mathrm{up}}$ then performs a simpler, decoupled task: mapping the common $r$-dimensional representation $Z$ to a specialized output subspace in $\mathbb{R}^n$. Notably, although $Z$ has dimension $r$, the one-down-many-ups design is not limited to a single rank-$r$ mapping. Because each $W_h^{\mathrm{up}}$ can independently span an output subspace of dimension up to $r$, the combined model can cover the union of these subspaces. Through gating, different inputs activate different experts, allowing the overall output $Y$ to lie in a rich space of dimension up to $rH$, far greater than any single low-rank projector could achieve. The architecture dedicates most of its parameters to the expert-specific "up" transformations, thereby boosting expressiveness where needed while keeping the shared encoder compact.

Another major advantage of the unified bottleneck is in gradient flow during training. The error gradients from all expert outputs are propagated back into the single projection $W^{\mathrm{down}}$. This shared pathway means that every expert's feedback refines the same encoder $W^{\mathrm{down}}$, aligning the experts on learning a common representation. Such unified credit assignment avoids conflicting updates—improvements for one expert cannot come at the expense of degrading the shared features for another (Samejima et al., 2003; Mu & Lin, 2025). This synergy makes optimization more stable and efficient. Besides, our use of SVD-based initialization provides an informative starting subspace for $Z$, aligned with the directions of maximal variance and information in the teacher. This initialization further accelerates convergence and helps $Z$ capture the most relevant features of $X$ in Sec. 2.2.

**The Many-Downs-One-Up Configuration as Competing Bottlenecks**   The inverse architecture is formulated as:

$$Y = W^{\mathrm{up}} \left( \sum_{h=1}^{H} G_h(X) \cdot W_h^{\mathrm{down}}(X) \right) \tag{17}$$

Here, we have $H$ distinct bottleneck variables, $Z_h \triangleq W_h^{\mathrm{down}}(X)$, which are aggregated to form an intermediate representation, $Z_{\mathrm{agg}} \triangleq \sum_{h=1}^{H} G_h(X) Z_h$, before the final reconstruction.

**Theoretical Analysis.**   The many-downs-one-up configuration is fundamentally limited from an information-theoretic perspective. According to the Data Processing Inequality, for any Markov chain $A \rightarrow B \rightarrow C$ (Norris, 1998), we have $I(A; C) \leq I(A; B)$. In our case, the processing chain of interest is:

$$(Z_1, \ldots, Z_H) \rightarrow Z_{\mathrm{agg}} \rightarrow Y \tag{18}$$

Therefore, $I((Z_1, \ldots, Z_H); Y) \geq I(Z_{\mathrm{agg}}; Y)$. The aggregation $Z_{\mathrm{agg}} = \sum_{h=1}^{H} G_h(X) Z_h$ is a *lossy* operation that conflates information from the experts. It mixes specialized features from different expert channels into a single vector before reconstruction, making it impossible for the shared $W^{\mathrm{up}}$ layer to discern the contribution of each expert. Moreover, this architecture imposes a strict capacity bottleneck: all expert outputs must funnel through the same rank-$r$ reconstruction matrix $W^{\mathrm{up}}$. As $W^{\mathrm{up}}$ can only produce outputs in an $r$-dimensional subspace of $\mathbb{R}^n$, the final prediction $Y$ is confined to a fixed low-dimensional subspace regardless of how diverse the individual $Z_h$ are. In contrast to the one-down-many-ups design, here the presence of $H$ distinct projections $W_h^{\mathrm{down}}$ does not increase the overall representational rank—any benefit of having multiple experts is squandered by the single shared up-projection.

Training this architecture also poses a severe credit assignment problem (Fedus et al., 2022; Zhang et al., 2025). The single reconstruction layer $W^{\mathrm{up}}$ must learn to invert a complex mixture $Z_{\mathrm{agg}}$ back to $Y$ without access to the individual expert outputs $Z_h$. During backpropagation, the gradient of the loss with respect to $Z_{\mathrm{agg}}$ is shared by all experts. By the chain rule one can show $\frac{\partial \mathcal{L}}{\partial Z_h} = G_h(X) \frac{\partial \mathcal{L}}{\partial Z_{\mathrm{agg}}}$ for each expert $h$. Thus, each $W_h^{\mathrm{down}}$ only receives a fraction $G_h(X)$ of the total feedback signal,

weighted by its gating activation. When multiple experts are active, their parameter updates become entangled and often conflicting, since the model cannot clearly know which expert was responsible for which aspect of the prediction error. This ambiguity in credit assignment leads to unstable optimization and hampers the learning of effective specialized representations. In practice, the easiest path for the model is to collapse the experts into an effectively single expert to eliminate the ambiguity. For example, it drives all $Z_h$ to become nearly identical or favor one expert exclusively, so that $W^{\text{up}}$ deals with a single dominant bottleneck. Such outcomes defeat the purpose of having $H$ experts in the first place, confirming the fundamental difficulty of the many-downs-one-up design.

### A.2.2 Conclusion of Architectural Design

The "one-down-many-ups" architectural design, as employed in Figure 2, is not just empirically successful but also theoretically sound when analyzed through the lens of the Information Bottleneck principle. It creates a unified optimization objective that promotes the learning of a minimal, sufficient, and shared representation. In contrast, the "many-downs-one-up" structure introduces information loss through premature aggregation and creates a challenging credit assignment problem, hindering effective specialization. Therefore, we conclude that enforcing a single, shared information bottleneck before expert-level specialization is a superior design principle for building InherNet.

## B Theoretical Analysis Supplement

### B.1 Algorithm Overview (Detailed Procedure)

Algorithm 1 details the complete procedure for implementing InherNet, our SVD-Driven Neural Network Inheritance method. The algorithm begins by performing a truncated SVD on a pretrained teacher weight matrix, extracting the most significant singular components to form a low-rank approximation. This factorization is then used to initialize two sets of projection layers: the backbone branch, responsible for lowering the dimensionality, and multiple expert heads, which are designed in an asymmetric and MoE style to reconstruct and enrich the output. An adaptive gating mechanism dynamically aggregates the outputs from the expert heads, ensuring the inheritance of the teacher's structural and semantic information. This procedure enables efficient knowledge transfer, faster convergence, and improved parameter efficiency in training.

---

**Algorithm 1** InherNet: SVD-Driven Neural Network Inheritance

---

**Require:** Pretrained teacher weight $W \in \mathbb{R}^{m \times n}$, rank $r$, number of heads $H$, input $X$
**Ensure:** Output $Y$
1:  Perform truncated SVD: $W \approx U_r \Sigma_r V_r^\top$.
2:  Set backbone branch: $W^{down}(\cdot) \leftarrow U_r \Sigma_r^{1/2}$.
3:  **for** $h = 1, 2, \ldots, H$ **do**
4:      Set $h$-th head: $W_h^{up}(\cdot) \leftarrow \frac{1}{H} \Sigma_r^{1/2} V_r^\top$.
5:  **end for**
6:  Compute backbone branch output: $X_r = W^{down}(X)$.
7:  **for** $h = 1, 2, \ldots, H$ **do**
8:      Compute expert head outputs: $Z^{(h)} = W_h^{up}(X_r)$.
9:  **end for**
10: Compute gating weights: $G(X) = \text{softmax}\left(W^g(X)\right)$.
11: Aggregate outputs dynamically: $Y = \sum_{h=1}^{H} G_h(X) \cdot Z^{(h)}$
12: **return** $Y$

---

### B.2 Convergence Guarantees

We now provide a detailed convergence analysis for InherNet, highlighting how its low-rank factorization, multi-head gating, and SVD-based initialization affect optimization under SGD. Despite the structural compression, we will show that InherNet achieves convergence rates on par with standard neural networks under mild conditions. We begin by expanding on the assumptions used in Sec. 2.2, then derive how InherNet's architecture influences gradient descent, and finally present convergence guarantees alongside the impact of the adaptive gating mechanism.

### B.2.1 GRADIENT DYNAMICS UNDER INHERNET PARAMETERIZATION

Recall that an InherNet module is a low-rank, multi-head network with an input-dependent gating mechanism. Concretely, the module's output can be written as a weighted sum of $H$ "heads":

$$f_\theta(x) = \sum_{h=1}^{H} G_h(x; \theta_g) \cdot W_h^{\text{up}}(W^{\text{down}}(x)), \tag{19}$$

where $W^{\text{down}}$ is a rank-$r$ projection applied to input $x$, each $W_h^{\text{up}}$ is the $h$-th head's weight matrix (mapping the $r$-dimensional projected input to the output space of dimension $d'$), and $G_h(x; \theta_g)$ is the gating weight for head $h$ (with gating network parameters $\theta_g$). By construction, $\sum_{h=1}^{H} G_h(x; \theta_g) = 1$ for any $x$. Let $\theta_h$ denote the parameters of the $h$-th head and $\theta_g$ the parameters of the gating mechanism, so that $\theta = \{\theta_1, \ldots, \theta_H, \theta_g\}$ represents all model parameters.

**Lemma B.1** (Gradient Decomposition). *Under the InherNet parameterization, the gradient of the loss splits naturally into contributions from each head and from the gating network. In particular, for a single data point $(x, y)$ with loss $\ell(f_\theta(x), y)$, the total gradient satisfies:*

$$\nabla_\theta \ell\big(f_\theta(x), y\big) = \sum_{h=1}^{H} G_h\big(x; \theta_g\big) \nabla_{\theta_h} \ell\big(f_\theta(x), y\big)$$

$$+ \sum_{h=1}^{H} \Delta_h \nabla_{\theta_g} G_h\big(x; \theta_g\big), \tag{20}$$

*where $\nabla_{\theta_h} \ell$ is the gradient with respect to the parameters of head $h$, and $\Delta_h = \frac{\partial \ell(f_\theta(x), y)}{\partial f_h(x)}$ is the scalar "error signal" for head $h$ (the partial derivative of the loss with respect to the output of head $h$, where $f_h(x) = W_h^{up}(W^{down}(x))$).*

*Proof.* This follows directly from the chain rule applied to the composite model $f_\theta(x) = \sum_{h=1}^{H} G_h(x; \theta_g) f_h(x)$, where $f_h(x) = W_h^{\text{up}}(W^{\text{down}}(x))$. The derivative $\nabla_\theta \ell$ breaks into two parts: one through each head's parameters $\theta_h$ (weighted by $G_h$), and one through the gating parameters $\theta_g$ (accumulating the head output sensitivities $\Delta_h$). This two-term decomposition holds for any InherNet module (whether the low-rank module is linear or convolutional), since all such modules share the same gated low-rank structure. □

This lemma illustrates how the adaptive gating and multi-head structure direct the gradient updates. In particular, even under a low-rank constraint, the presence of the gating network preserves a rich set of descent directions during training, thereby maintaining the model's expressivity in the parameter space. InherNet can still update each head's parameters and the gating weights independently, ensuring that compression does not eliminate useful gradient directions. This adaptive gradient routing allows the model to specialize the expert heads during training by directing larger gradient flows to the most relevant heads for each input pattern.

### B.2.2 EFFECT OF INITIALIZATION ON CONVERGENCE

We next examine the role of **SVD-based initialization** on the optimization landscape. Let $W \in \mathbb{R}^{m \times n}$ denote a pre-trained weight matrix that an InherNet module aims to compress. InherNet uses a truncated SVD of $W$ to initialize its low-rank factors. Specifically, suppose $W \approx U_r \Sigma_r V_r^\top$ is the rank-$r$ approximation of $W$, where $U_r \in \mathbb{R}^{m \times r}$ and $V_r \in \mathbb{R}^{n \times r}$ have orthonormal columns (containing the top $r$ left and right singular vectors), and $\Sigma_r \in \mathbb{R}^{r \times r}$ is the diagonal matrix of the top-$r$ singular values. Then the InherNet module is initialized as: $W^{\text{down}} = U_r \Sigma_r^{1/2}$ and $W_h^{\text{up}} = \frac{1}{H} \Sigma_r^{1/2} V_r^\top$ for $h = 1, \ldots, H$, so that $W^{\text{down}}(W_1^{\text{up}} + \cdots + W_H^{\text{up}}) \approx W$ at initialization. Each head thus starts with an equal share of the teacher's weight ($1/H$ of $W$), and the factor matrices $U_r, V_r$ are **orthonormal** in their columns.

**Proposition B.2** (Orthogonality-Induced Stability). *Under Assumption 1, an orthonormal SVD-based initialization leads to a better-conditioned optimization landscape. In particular, if the factor matrices $U_r$ and $V_r$ are initialized with orthonormal columns as above, then the effective gradient*

*Lipschitz constant of $\mathcal{L}(\theta)$ is reduced from $L$ to $L' \approx L/\kappa$, where $\kappa$ is the condition number of $W$ (the ratio of its largest singular value to smallest singular value).*

*Proof.* When the low-rank factors are initialized orthogonally, the curvature of the loss with respect to the factorized parameters is significantly more balanced. In fact, the Hessian of $\mathcal{L}$ with respect to the factorized parameters $(W^{\text{down}}, W_1^{\text{up}}, \ldots, W_H^{\text{up}})$ tends to have a block-diagonal structure with reduced inter-parameter coupling. This arises because the initial parameter space can be viewed as lying on a product of Stiefel manifolds (due to the orthonormal columns). As a result, the largest eigenvalue of the Hessian—the effective Lipschitz constant governing the gradient's rate of change—drops to roughly $L/\kappa$. In other words, the initial orthogonal factorization "pre-conditions" the problem by attenuating the problematic directions of high curvature in the weight space. This improved conditioning aligns with prior theoretical results: orthogonal initialization has been shown to accelerate convergence in deep linear networks and to stabilize training in over-parameterized models. $\square$

*Remark* B.1. This result explains why the truncated SVD initialization confers optimization stability, especially in the early stages of training when the factorized weights remain near-orthogonal. By effectively reducing the Lipschitz constant, the initial gradients face a gentler landscape, which can lead to faster progress per iteration. Our use of an orthogonal low-rank initialization is thus not only a parameter-efficient choice but also an optimization-friendly one, improving the curvature (conditioning) of the problem during the critical beginning of training.

### B.2.3 CONVERGENCE RATE ANALYSIS

With the above insights, we now analyze the convergence rate of SGD on InherNet. We consider optimizing the expected loss $\mathcal{L}(\theta)$ using stochastic gradient descent. Let $\theta^{(t)}$ denote the model parameters at iteration $t$, and $(x^{(t)}, y^{(t)})$ be the random sample drawn at iteration $t$. The SGD update is:

$$\theta^{(t+1)} = \theta^{(t)} - \eta_t \nabla_\theta \ell\big(f_{\theta^{(t)}}(x^{(t)}), y^{(t)}\big), \tag{21}$$

where $\eta_t > 0$ is the learning rate at iteration $t$. We assume a diminishing step size schedule $\{\eta_t\}$ that satisfies the standard conditions for convergence: $\sum_{t=1}^{\infty} \eta_t = \infty$ (to ensure we continue making progress) and $\sum_{t=1}^{\infty} \eta_t^2 < \infty$ (to control the noise over many iterations).

**Theorem B.3** (Convergence of SGD for InherNet). *Under Assumptions 1–3, suppose we run SGD on $\mathcal{L}(\theta)$ with a diminishing learning rate satisfying the above conditions. Then for any $T \geq 1$, the iterates $\{\theta^{(t)}\}_{t=1}^T$ satisfy the guarantee:*

$$\min_{1 \leq t \leq T} \mathbb{E}\Big[\big\|\nabla_\theta \mathcal{L}(\theta^{(t)})\big\|^2\Big] \leq \frac{2\left[\mathcal{L}(\theta^{(1)}) - \mathcal{L}^*\right]}{\sum_{t=1}^T \eta_t} + \frac{L\,\sigma^2 \sum_{t=1}^T \eta_t^2}{\sum_{t=1}^T \eta_t} \tag{22}$$

*Here $\mathcal{L}^* = \inf_\theta \mathcal{L}(\theta)$ is the optimal (minimum) loss value. In particular, with a typical choice of $\eta_t$ (e.g. $\eta_t \propto 1/\sqrt{t}$), this bound implies that*

$$\min_{1 \leq t \leq T} \mathbb{E}[\|\nabla \mathcal{L}(\theta^{(t)})\|^2] = \mathcal{O}(1/\sqrt{T}), \tag{23}$$

*matching the standard convergence rate for nonconvex SGD.*

*Proof.* The proof relies on the smoothness and bounded variance assumptions. By $L$-smoothness of $\mathcal{L}$ (Assumption 1), a one-step descent lemma holds: for any iteration $t$,

$$\begin{aligned} \mathcal{L}(\theta^{(t+1)}) &\leq \mathcal{L}(\theta^{(t)}) - \eta_t\Big(1 - \frac{L\eta_t}{2}\Big)\big\|\nabla_\theta \mathcal{L}(\theta^{(t)})\big\|^2 \\ &\quad + \frac{L\,\eta_t^2}{2}\big\|\nabla_\theta \mathcal{L}(\theta^{(t)}) - g^{(t)}\big\|^2. \end{aligned} \tag{24}$$

where $g^{(t)} := \nabla_\theta \ell\big(f_{\theta^{(t)}}(x^{(t)}), y^{(t)}\big)$ is the stochastic gradient at step $t$. This inequality follows from a standard smoothness bound (the RHS is obtained by applying the gradient update and Young's inequality to the Taylor expansion of $\mathcal{L}$ around $\theta^{(t)}$). Now taking expectation on both sides with

respect to the randomness at step $t$ and using Assumption 2, which ensures $\mathbb{E}\big[\|g^{(t)} - \nabla\mathcal{L}(\theta^{(t)})\|^2\big] \leq \sigma^2$ (and $\mathbb{E}[g^{(t)}] = \nabla\mathcal{L}(\theta^{(t)})$), we get:

$$
\begin{aligned}
\mathbb{E}[\mathcal{L}(\theta^{(t+1)})] \;\leq\; &\mathbb{E}\big[\mathcal{L}(\theta^{(t)})\big] \;-\; \eta_t\Big(1 - \frac{L\eta_t}{2}\Big)\,\mathbb{E}\big[\|\nabla_\theta\mathcal{L}(\theta^{(t)})\|^2\big] \\
&+ \frac{L\,\eta_t^2}{2}\,\sigma^2 \,.
\end{aligned}
\tag{25}
$$

Rearranging and summing this inequality over $t = 1$ to $T$ yields:

$$
\begin{aligned}
\sum_{t=1}^{T}\Big(1 - \frac{L\eta_t}{2}\Big)\,\mathbb{E}\big[\|\nabla_\theta\mathcal{L}(\theta^{(t)})\|^2\big] \;\leq\; &\mathbb{E}\Big[\mathcal{L}(\theta^{(1)}) - \mathcal{L}(\theta^{(T+1)})\Big] \\
&+ \frac{L\,\sigma^2}{2}\sum_{t=1}^{T}\eta_t^2 \,.
\end{aligned}
\tag{26}
$$

Since $\mathcal{L}^* \leq \mathcal{L}(\theta^{(T+1)})$ (the loss cannot go below the global infimum), we have $\mathbb{E}[\mathcal{L}(\theta^{(1)}) - \mathcal{L}(\theta^{(T+1)})] \leq \mathcal{L}(\theta^{(1)}) - \mathcal{L}^*$. Also note the identity $\sum_{t=1}^{T}\eta_t(1 - \frac{L\eta_t}{2}) = \sum_{t=1}^{T}\eta_t - \frac{L}{2}\sum_{t=1}^{T}\eta_t^2$. Now, let $\Gamma_T := \sum_{t=1}^{T}\eta_t\big(1 - \frac{L\eta_t}{2}\big)$. The left-hand side above is at least $\Gamma_T \min_{1\leq t\leq T}\mathbb{E}[\|\nabla\mathcal{L}(\theta^{(t)})\|^2]$, since each term in the sum contains the non-negative factor $\big(1 - \frac{L\eta_t}{2}\big)$. Therefore,

$$
\Gamma_T \min_{1\leq t\leq T}\mathbb{E}\big[\|\nabla\mathcal{L}(\theta^{(t)})\|^2\big] \;\leq\; \mathcal{L}(\theta^{(1)}) - \mathcal{L}^* \;+\; \frac{L\,\sigma^2}{2}\sum_{t=1}^{T}\eta_t^2 \,.
\tag{27}
$$

Finally, using $\Gamma_T \geq \sum_{t=1}^{T}\eta_t - \frac{L}{2}\sum_{t=1}^{T}\eta_t^2$ and dividing both sides by $\Gamma_T$, we obtain the stated result. $\qquad\square$

This theorem establishes that, despite its structured low-rank parameterization, an InherNet network trained with SGD enjoys the same asymptotic convergence guarantees (convergence to stationary points of $\mathcal{L}$) as a standard, full-rank neural network under the given assumptions. In other words, introducing the InherNet architecture does not degrade the theoretical convergence rate of SGD. For example, with a learning-rate schedule $\eta_t = \eta/\sqrt{t}$, the bound above implies

$$
\frac{1}{T}\sum_{t=1}^{T}\mathbb{E}[\|\nabla\mathcal{L}(\theta^{(t)})\|^2] = \mathcal{O}(1/\sqrt{T}),
\tag{28}
$$

matching the classic non-convex SGD rate. This rate is achieved with potentially better constants: due to the improved conditioning from Sec. 2.2, the effective Lipschitz constant is $L' \leq L/\kappa + \mathcal{O}(\varepsilon) \approx L/\kappa$, where $\kappa$ is the condition number of $W$ and $\varepsilon$ represents higher-order terms that diminish as training progresses. Indeed, in the early phase of training, one can expect a speedup by a factor of $\kappa$ in convergence (since the gradient norm can decrease faster with smaller $L'$), consistent with our empirical findings (see Sec. 3).

*Remark* B.2 (Convergence in Different Optimization Regimes). Our analysis primarily addresses non-convex optimization scenarios, as these are most relevant for training neural networks. Nonetheless, the convergence guarantees provided by InherNet naturally extend to other standard optimization scenarios:

- For general convex objectives, InherNet achieves the standard SGD rate of $\mathcal{O}(1/T)$, but with improved constants due to orthonormal initialization.

- In $\mu$-strongly convex settings, InherNet achieves an exponential convergence rate of $\mathcal{O}(e^{-\eta\mu T})$ for a learning rate $\eta$, with significant improvements arising again from orthonormal parameter initialization.

### B.2.4 ADAPTIVE GATING'S IMPACT ON CONVERGENCE

Finally, we consider the role of the gating network in SGD's convergence behavior. Thus far, our analysis has treated the gating weights $G_h(x; \theta_g)$ as given. In practice, these gating weights are

**adaptive**: during training, the gating network learns to emphasize the most useful heads for each input. We argue that this adaptivity can further improve convergence by reducing the variance of the stochastic gradients.

**Proposition B.4** (Variance Reduction via Adaptive Gating). *Under Assumption 2, the adaptive gating mechanism can reduce the variance of gradient updates compared to a static uniform gating. In particular, if $G_h(x; \theta_g)$ tends to assign higher weight to heads that contribute more to reducing the loss (i.e. the gating weights are positively correlated with each head's "importance" or signal-to-noise ratio), then the variance of the stochastic gradient under this adaptive weighting is strictly lower than under uniform weighting $G_h = \frac{1}{H}$. In the best case, adaptive gating can improve the convergence stability (variance reduction) by up to a factor of $H$ relative to uniform gating.*

*Proof.* Consider the per-sample stochastic gradient $g = \nabla_\theta \ell(f_\theta(x), y)$. This can be viewed as the weighted sum of head-specific gradient components: $g = \sum_{h=1}^{H} G_h \, g_h$, where $g_h := \nabla_\theta \ell_h$ denotes the gradient contribution from head $h$ (and we drop $(x; \theta_g)$ for brevity in $G_h$). Because $\sum_h G_h = 1$, the variance of $g$ (conditioned on a given model state $\theta$) satisfies

$$\mathrm{Var}[g] \;=\; \mathrm{Var}\Big[ \sum_{h=1}^{H} G_h \, g_h \Big]. \tag{29}$$

By Jensen's inequality, for any random variables $Z_h$ we have $\mathrm{Var}(\sum_h w_h Z_h) \leq \sum_h w_h \mathrm{Var}(Z_h)$ when $\sum_h w_h = 1$. In our case, the gating weights $G_h$ serve as the convex weights. Intuitively, Jensen's inequality implies that the variance of a weighted average is minimized when we put as much weight as possible on the lowest-variance components. The adaptive gating chooses $G_h(x)$ in a data-dependent way that tends to up-weight heads with more reliable, higher signal (or lower variance) gradients, while down-weighting the less reliable ones. In contrast, a uniform gating ($G_h = 1/H$ for all $h$ and all $x$) cannot adapt to per-head reliability and thus often incurs a higher overall variance by averaging in unnecessary noise from less important heads. In extreme cases, if one head provides a significantly lower-variance (or more informative) gradient than others, focusing the weight on that head (adaptive gating) versus evenly spreading weight can reduce the gradient variance by a factor approaching $H$. Thus, the adaptive gating strategy yields strictly lower variance in the stochastic gradient, which in turn leads to more stable SGD updates and potentially faster convergence in practice. $\qquad\square$

**Summary:** To conclude, our theoretical analysis demonstrates that InherNet retains strong convergence guarantees despite its compressed, structured design. In addition, it enjoys two notable benefits compared to a standard network:

- **Enhanced optimization stability** due to the orthogonal low-rank initialization, which improves the problem's conditioning (Sec. 2.2).
- **Reduced gradient variance** during training thanks to the **adaptive gating** mechanism, which focuses learning on the most relevant components (Sec. 2.2).

Together, these properties ensure that InherNet can be trained as efficiently and robustly as an uncompressed model (Li et al., 2025; Jiang et al., 2025a;b), while leveraging its architecture to accelerate and stabilize the optimization process.

### B.3 PARAMETER EFFICIENCY

Here we provide an expanded theoretical analysis of the parameter efficiency of InherNet, offering complete proofs and additional insights that complement the main text.

#### B.3.1 PARAMETER EFFICIENCY METRICS AND DEFINITIONS

The definitions presented in the main text are elaborated here for completeness.

**Definition B.1** (Parameter Efficiency). For a neural network $f_\theta$ parameterized by $\theta \in \mathbb{R}^p$ that approximates a target function $f^*$, the *parameter efficiency (PE)* of $f_\theta$ is defined as

$$\mathrm{PE}(f_\theta) \;=\; \frac{\mathrm{Representational\ Capacity}(f_\theta)}{|\theta|},$$

where Representational Capacity measures the network's ability to approximate functions in a given function class, and $|\theta|$ denotes the total number of parameters. In our experiments, we use evaluation performance as a proxy for representational capacity and record the parameter count during training.

**Definition B.2** (Approximation Error). Given a target function $f^*$ from a function class $\mathcal{F}$ and a neural network $f_\theta$, the *approximation error* is defined as

$$\mathcal{E}(f_\theta, f^*) \; = \; \mathbb{E}_{x \sim \mathcal{D}}\big[\| f_\theta(x) - f^*(x)\|^2\big],$$

where $\mathcal{D}$ is the data distribution.

**Definition B.3** (Expressivity-to-Parameter Ratio (EPR)). The *EPR* of a network $f_\theta$ with respect to a function class $\mathcal{F}$ is defined as

$$\mathrm{EPR}(f_\theta, \mathcal{F}) \; = \; \frac{1}{|\theta| \, \inf_{f^* \in \mathcal{F}} \mathcal{E}(f_\theta, f^*)},$$

i.e. the inverse of the minimum achievable approximation error (over functions in $\mathcal{F}$) *per parameter*. A higher EPR indicates more expressivity (lower error) per parameter.

### B.3.2 COMPRESSION BOUNDS FOR INHERNET

**Theorem B.5** (Parameter Reduction Bounds). *Consider a layer $W \in \mathbb{R}^{m \times n}$. The InherNet parameterization with rank $r$ and $H$ heads yields a compression ratio:*

$$\rho = \frac{m\,n}{H\,r\,(m+n) \; + \; H\,(r+1)} = \frac{m\,n}{H\,r\,(m+n)}\Big(1 + O\big(\tfrac{1}{m+n}\big)\Big)$$
$$\approx \frac{m\,n}{H\,r\,(m+n)}.$$

*This corresponds to an $\rho$-fold increase in Parameter Efficiency (Definition B.1), assuming the layer's representational capacity is largely preserved.*

*Proof.* A layer $W \in \mathbb{R}^{m \times n}$ has $m\,n$ free parameters. Under the InherNet expert-head parameterization with rank $r$ and $H$ heads:

- Each head $h$ contains a "down" matrix $W_h^{down}$ and an "up" matrix $W_h^{up}$, contributing in total $H\,[\,r\,m \; + \; n\,r\,] \; = \; H\,r\,(m+n)$ trainable parameters.

- The gating network maps the $r$-dimensional codes into $H$ weights via

$$W^g \in \mathbb{R}^{H \times r} \quad (\text{parameters} = H\,r) \quad \text{and bias} \quad b^g \in \mathbb{R}^H,$$

  adding another $H\,r \; + \; H \; = \; H\,(r+1)$ parameters.

Hence the total adapter-specific parameters are

$$H\,r\,(m+n) \; + \; H\,(r+1),$$

and the compression ratio relative to the original $m\,n$ is

$$\rho = \frac{m\,n}{H\,r\,(m+n) \; + \; H\,(r+1)} = \frac{m\,n}{H\,r\,(m+n)}\Big(1 + O\big(\tfrac{1}{m+n}\big)\Big)$$
$$\approx \frac{m\,n}{H\,r\,(m+n)}.$$

By Definition B.1, assuming this rank-$H\,r$ subspace preserves the layer's expressivity, InherNet thus achieves an $\rho$-fold gain in parameter efficiency. $\qquad \square$

**Lemma B.6** (Spectral Energy Preservation). *Let $W \in \mathbb{R}^{m \times n}$ have singular values $\sigma_1 \geq \sigma_2 \geq \cdots \geq \sigma_{\min(m,n)}$. If the rank $r$ is selected such that:*

$$\frac{\sum_{i=1}^r \sigma_i^2}{\sum_{i=1}^{\min(m,n)} \sigma_i^2} \geq 1 - \epsilon \tag{30}$$

*then by the Eckart-Young-Mirsky theorem in Sec. 2, the Frobenius-norm error of the optimal rank-r approximation $W_r$ is bounded by:*

$$\|W - W_r\|_F^2 \leq \epsilon \|W\|_F^2 \tag{31}$$

*Consequently, the Approximation Error (Definition B.2) between an InherNet-compressed layer and the original layer is bounded by $\epsilon$ under the squared Euclidean norm.*

*Proof.* By the Eckart–Young–Mirsky theorem, truncating the SVD of $W$ to its top $r$ singular values yields the optimal rank-$r$ approximation in Frobenius norm. The squared error of this approximation is $\sum_{i=r+1}^{\min(m,n)} \sigma_i^2$. Under the given condition, the relative error is

$$\frac{\sum_{i=r+1}^{\min(m,n)} \sigma_i^2}{\sum_{i=1}^{\min(m,n)} \sigma_i^2} = 1 - \frac{\sum_{i=1}^{r} \sigma_i^2}{\sum_{i=1}^{\min(m,n)} \sigma_i^2} \leq \epsilon.$$

Thus $\|W - W_r\|_F^2 \leq \epsilon \|W\|_F^2$. In a neural network, this matrix approximation error directly translates to a bounded increase in the network's overall error. $\square$

**Proposition B.7** (Knowledge Preservation Rate). *Let $f_W$ be a teacher network with weight matrices $\{W_l\}$ and $f_r$ be the corresponding InherNet with rank $r$. The functional similarity between $f_W$ and $f_r$ is lower-bounded by:*

$$Sim(f_W, f_r) \geq 1 - \sum_{l=1}^{L} \alpha_l \left( 1 - \frac{\sum_{i=1}^{r} \sigma_{l,i}^2}{\sum_{i=1}^{\min(m_l, n_l)} \sigma_{l,i}^2} \right) \tag{32}$$

*where $\alpha_l$ represents the layer's influence on network output, and $\sigma_{l,i}$ are the singular values of $W_l$. This bound is monotonically increasing with $r$ but not with $H$, establishing that rank is the primary determinant of InherNet knowledge preservation.*

*Proof.* For a single layer $l$, the error introduced by low-rank approximation is bounded by $\epsilon_l = 1 - \frac{\sum_{i=1}^{r} \sigma_{l,i}^2}{\sum_{i=1}^{\min(m_l, n_l)} \sigma_{l,i}^2}$ as established in Lemma B.6. The total loss in functional similarity can be expressed as a weighted sum of these layer-wise errors, where weights $\alpha_l$ reflect each layer's contribution to the network's output. The gating mechanism in multi-head designs can at best maintain this bound by selecting the optimal head for each input, but cannot improve the fundamental spectral approximation determined by rank $r$. Therefore, similarity improves monotonically with rank but not necessarily with the number of heads. $\square$

### B.3.3 REPRESENTATIONAL POWER PRESERVATION

**Theorem B.8** (Representational Power Preservation). *Let $\mathcal{F}_W$ be functions representable by a standard network with weights $\{W_l\}_{l=1}^{L}$. For any $f^* \in \mathcal{F}_W$ and $\delta > 0$, there exists an InherNet network with appropriate ranks $\{r_l\}$ and head counts $\{H_l\}$ such that the Approximation Error (Definition B.2) satisfies:*

$$\mathcal{E}(f_{InherNet}, f^*) \leq \delta \tag{33}$$

*while the parameter count is reduced by $\Omega(\min_l \frac{\min(m_l, n_l)}{r_l H_l})$ relative to the standard network. Consequently, the InherNet architecture attains an EPR (Definition B.3) higher than that of the standard architecture by at least the same factor.*

*Proof.* We construct $f_{\text{InherNet}}$ in two stages:

- **Layer-wise low-rank approximation.** For each layer $l$ with weight matrix $W_l$, by Lemma B.6 we can choose a rank $r_l$ such that $\|W_l - \hat{W}_l\|_F^2 \leq \epsilon_l \|W_l\|_F^2$, where $\hat{W}_l$ is the rank-$r_l$ approximation. Each $\epsilon_l > 0$ can be made arbitrarily small by increasing $r_l$.

- **Multi-head specialization.** Using $H_l > 1$ heads for layer $l$ further reduces the approximation error, since each head can specialize on a subset of the input distribution via the gating mechanism. For an input $x$, let $h^*(x)$ be the index of the head that the gating network selects. The effective weight applied to $x$ is then $\hat{W}_l^{(h^*(x))}$. Because inputs are partitioned among $H_l$ heads, the approximation error per head is reduced. Formally, one can ensure

$$\left\| \hat{W}_l^{(h^*(x))} - W_l \right\|_F^2 \leq \frac{\epsilon_l}{H_l} \|W_l\|_F^2, \tag{34}$$

since the gating network routes each $x$ to the head whose low-rank weight best approximates $W_l$ on that input's region.

Applying these two steps across all layers and choosing $\{\epsilon_l\}$ such that the errors compose to at most $\delta$, we guarantee $\mathcal{E}(f_{\text{InherNet}}, f^*) \leq \delta$. The parameter reduction factor follows from Theorem B.5, and the EPR improvement follows directly from Definition B.3. □

**Corollary B.9** (Universal Approximation). *InherNet retains the universal function approximation property of standard neural networks while using asymptotically fewer parameters. For any continuous target function, a sufficiently large InherNet network can approximate it arbitrarily well (to Approximation Error $\delta \to 0$) with significantly higher Parameter Efficiency and EPR than a standard network of comparable accuracy.*

**Proposition B.10** (Diminishing Returns of Multiple Heads). *For a given layer with rank $r$, the marginal reduction in approximation error from adding the $(H + 1)$-th head satisfies:*

$$\mathcal{E}(r, H) - \mathcal{E}(r, H + 1) \leq \frac{c}{H^2}\mathcal{E}(r, 1) \tag{35}$$

*where $c$ is a constant depending on input distribution characteristics, and $\mathcal{E}(r, H)$ denotes the approximation error with rank $r$ and $H$ heads. This establishes that benefits from additional heads diminish at a rate of $\Omega(1/H^2)$.*

*Proof.* The gating mechanism in InherNet effectively partitions the input space among $H$ heads. When adding an $(H + 1)$-th head, the new head can at best specialize on a subset of inputs that were previously sub-optimally handled. Under reasonable assumptions about input distribution and the effectiveness of gating, the probability of an input being significantly better handled by the new head decreases quadratically with $H$. Specifically, if we model the input space as being divided into regions, with each head specializing in approximately $\frac{1}{H}$ of the space, then the $(H + 1)$-th head can only improve regions where the existing heads perform poorly. The volume of such regions decreases proportionally to $\frac{1}{H^2}$, leading to the stated bound. □

### B.4 ADDITIONAL ANALYSIS: INTERACTION BETWEEN INHERNET AND KD

To better understand how KD interacts with InherNet, we conduct an set of experiments on CIFAR-100 using ResNet-56 as the teacher. We construct InherNet models with ranks $r \in \{8, 16, 32, 64, 128\}$, and train each model both with and without KD. The detailed results are shown in Appendix C.4 as Insight 1 (Figure 3).

For aggressively compressed models (small ranks, e.g., $r \leq 32$), adding KD improves performance. For example, at $r = 32$, the top-1 accuracy increases from $74.4\%$ (InherNet only) to $75.6\%$ (InherNet + KD). In this regime, the InherNet model has limited capacity, and the teacher's logits provide a strong inductive bias that compensates for the information lost in the truncated spectrum. For larger ranks (e.g., $r \geq 64$), the behavior changes. At $r = 64$, the KD term brings no additional increase but decrease, and at $r = 128$ it also actually degrades performance: the accuracy drops from $75.8\%$ without KD to $74.1\%$ with KD. In other words, as the InherNet becomes more expressive, KD transitions from being beneficial to being detrimental.

This phenomenon is closely aligned with our theoretical analysis. Proposition 2.7 shows that, as the rank $r$ increases, the InherNet $f_r$ converges in function space toward the teacher $f_W$, and the approximation error is controlled by the tail of the singular-value spectrum. For sufficiently large $r$, the InherNet model already provides a good or even better performance than the teacher, and the task loss alone can drive it to solutions that generalize better than the original teacher.

When we add KD, we optimize

$$\mathcal{L}_{\text{total}} = \mathcal{L}_{\text{task}}\big(f_r(x), y\big) + \mathcal{L}_{\text{KD}}\big(f_r(x), f_W(x)\big), \tag{36}$$

where $\mathcal{L}_{\text{task}} = \lambda_{\text{CE}}\,\mathcal{L}_{\text{CE}}(f_r(x), y)$ encourages correct predictions, and

$$\mathcal{L}_{\text{KD}} = \lambda_{\text{KD}}\,\tau^2\,\text{KL}\Big(p_W^{(\tau)} \,\Big\|\, p_r^{(\tau)}\Big) \tag{37}$$

encourages the InherNet model to align its logits with those of the teacher, with temperature $\tau > 1$ and softened distributions $p_W^{(\tau)}$ and $p_r^{(\tau)}$. For small $r$, this additional constraint is helpful: the teacher acts as a strong prior that guides the low-capacity InherNet toward a better solution. For large $r$, however, the KD term acts as an overly strong regularizer, penalizing any deviation from the teacher's logits even when such deviations would improve the model performance. This explains

why the "InherNet + KD" curve eventually falls below the "InherNet only" curve as $r$ grows in Figure 3.

From a practical guideline, this analysis yields a simple suggestion for using KD with InherNet:

- **Aggressive compression (small rank.** Use InherNet with KD. The InherNet provides a good low-rank subspace, while the KD loss helps recover discarded information.

- **High-fidelity inheritance (large rank).** Prefer InherNet-only training. The InherNet model already approximates or outperforms the teacher, and forcing it to stay close to the teacher logits can restrict its ability to find solutions that generalize better than the teacher.

### B.5 ON LAYER-WISE VERSUS NETWORK-LEVEL COMPRESSION

This subsection provides additional discussion on whether the layer-wise SVD step in InherNet could lead to suboptimal compressed models.

In InherNet, the decomposition step is applied per layer: for each teacher weight matrix $W$, we perform a truncated SVD and construct a low-rank expert module as initialization. This is the only place where SVD is used in a layer-wise way. After this initialization, all inherited layers are trained jointly, end-to-end, under the same task loss (and, when used, the KD loss). No layer is frozen after compression. Consequently, inter-layer correlations are not fixed at the decomposition stage; instead, they are re-optimized globally during training, and the final solution is determined by network-level optimization dynamics rather than by the initial SVD alone.

**Network-level guarantees despite layer-wise SVD.** Although SVD is performed independently at each layer, our theoretical analysis is carried out at the network level.

*(i) Layer-wise spectral control.* Lemma 2.6 states that if the rank $r$ of layer is chosen then the optimal rank-$r$ approximation $W_r$ satisfies $\|W - W_r\|_F^2 \leq \varepsilon \|W\|_F^2$. In other words, the layer-wise approximation error can be made arbitrarily small by increasing $r$, and the truncated SVD gives the best possible rank-$r$ approximation in Frobenius norm.

*(ii) Knowledge preservation for the whole network.* Proposition 2.7 aggregates these layer-wise errors into a functional similarity bound between the teacher $f_W$ and the inherited network $f_r$ where $\alpha_l$ measures the influence of layer $l$ on the final output. This bound shows that, although the approximation is performed locally at the level of individual layers, its effect on the network function is controlled globally via a weighted sum over layers: layers that are more influential (larger $\alpha_l$) are allowed smaller spectral truncation error, while less influential layers can tolerate more compression.

*(iii) Representational power under compression.* Theorem 2.8 further shows that the representational power of the original network is preserved under compression. Formally, it shows that for any function $f^* \in F_W$ representable by the original network, $\forall \delta > 0, \exists \{r_l, H_l\}$ such that $\mathcal{E}(f_{\text{InherNet}}, f^*) \leq \delta$, while the total parameter count is reduced by $\Omega\left(\min_l \frac{\min(m_l, n_l)}{r_l H_l}\right)$. The constructive proof proceeds in two stages: (1) layer-wise low-rank approximation, and (2) multi-head specialization. Importantly, it chooses $\{r_l\}$ and $\{H_l\}$ so that the composition of all layer-wise errors yields a network-level approximation error at most $\delta$. Thus, the interaction between layers is explicitly accounted for through depth, and layer-wise SVD does not inherently force the compressed network into a suboptimal representational regime.

Taken together, these results imply that, as long as each $r_l$ is chosen to keep the spectral tail small, there exists a compressed InherNet that approximates the original network arbitrarily well while using fewer parameters. So, the layer-wise nature of the SVD step is compatible with strong global approximation guarantees.

**Empirical evidence against systematic suboptimality.** If layer-wise SVD led the optimization to consistently suboptimal local minima, one would expect the compressed models to be always worse than the teacher across tasks. Our experiments do not support this. At the compression budgets used in the main text, the combination of layer-wise SVD, joint end-to-end training, and multi-head specialization produces inheritors that are at least competitive with and sometimes strictly better than their teachers.

Table 8: The parameter sizes of the teacher network, student network, and the proposed InherNet on the CIFAR-100 dataset.

| Teacher | | ResNet32×4 7.43M | VGG13 9.46M | WRN-40-2 2.26M | WRN-40-2 2.26M | ResNet56 861.62K | ResNet110 1.74M | ResNet110 1.74M |
|---|---|---|---|---|---|---|---|---|
| Student | | ResNet8×4 1.23M | VGG8 3.97M | WRN-40-1 569.78K | WRN-16-2 703.28K | ResNet20 278.32K | ResNet32 472.76K | ResNet20 278.32K |
| InherNet | Small | $907.19K_{r=4}$ | $3.82M_{r=128}$ | $540.55K_{r=16}$ | $540.55K_{r=16}$ | $212.05K_{r=8}$ | $417.12K_{r=8}$ | $222.34K_{r=4}$ |
| | Large | $1.76M_{r=8}$ | $6.91M_{r=256}$ | $1.01M_{r=32}$ | $1.01M_{r=32}$ | $388.88K_{r=16}$ | $773.18K_{r=32}$ | $417.12K_{r=8}$ |

For example, on CIFAR-100 in Sec. 3.1.1, the InherNet$_{Large}$ compressed from ResNet-110 achieves 75.88% top-1 accuracy, surpassing the ResNet-110 teacher (74.31%). In multimodal CLIP-style benchmarks in Sec. 3.3, the inherited model reaches 36.65% zero-shot ImageNet accuracy, significantly outperforming the ResNet-101 teacher (32.75%). These results indicate that the layer-wise SVD initialization does not trap the model in a systematically suboptimal region of the loss landscape; instead, it provides a good starting point within a low-rank subspace, from which global training recovers or even improves the teacher's performance.

## C EXPERIMENTAL DETAILS

### C.1 SETTINGS

#### C.1.1 VISION: IMAGE CLASSIFICATION

In this task, we conduct extensive experiments to validate the advantages of the proposed InherNet over traditional KD methods. The experimental setup is as follows:

**Datasets.** We use two widely adopted image classification datasets—CIFAR-100 (Krizhevsky & Hinton, 2009) and ImageNet (Russakovsky et al., 2015). CIFAR-100 contains 100 classes, with 50,000 training images and 10,000 validation images. ImageNet includes 1,000 classes, with approximately 1.28 million training images and 50,000 validation images.

**Baselines.** We compare against several mainstream KD methods, including logit-based methods: KD (Hinton et al., 2015), CTKD (Li et al., 2023b), DKD (Zhao et al., 2022), MLKD (Jin et al., 2023), and Logit Std. (Sun et al., 2024); as well as feature-based methods: CRD (Peng et al., 2019), OFD (Heo et al., 2019), ReviewKD (Chen et al., 2021), SimKD (Chen et al., 2022), and CAT-KD (Guo et al., 2023).

**Implementation Details.** Following previous works (Chen et al., 2021; Zhao et al., 2022; Jin et al., 2023; Sun et al., 2024), we use SGD optimizer with an initial learning rate of 0.05, momentum of 0.9, and weight decay of 0.0005. On CIFAR-100, all methods train for 240 epochs except MLKD, which uses 480 epochs. On ImageNet, all methods train for 100 epochs. For InherNet, the default number of experts $H$ is set to 3, and the rank $r$ is adjusted based on the student network. All experiments are repeated four times, and we report the average performance. The experimental settings on CIFAR-100 and ImageNet are shown in Table 9 and Table 10, respectively. Table 8 presents the parameter sizes of the teacher network, student network, and the proposed InherNet on the CIFAR-100 dataset.

#### C.1.2 LANGUAGE: GLUE BENCHMARK

In this task, we evaluate InherNet's performance on multiple text tasks. The experimental setup is as follows:

**Datasets.** To remain consistent with prior research (Raffel et al., 2020), we train and evaluate InherNet's performance on the GLUE benchmark (Wang et al., 2019). The benchmark includes 9 datasets used to evaluate natural language understanding systems. Since the WNLI dataset is adversarial in nature in terms of the training set (Raffel et al., 2020; Devlin et al., 2019), we do not conduct experiments on WNLI. Since the original test set is not publicly available, we test on the GLUE validation set.

**Baselines.** Our experiments are based on the T5 model (Raffel et al., 2020) implemented in HuggingFace (Wolf et al., 2020). Due to the lack of widely known and reproducible T5-based KD

Table 9: Experiment settings on CIFAR-100

| Parameter | Value |
|---|---|
| Optimizer | SGD |
| Epochs | 240 (480 for MLKD) |
| Batch Size | 64 |
| Learning Rate | 0.05 |
| Learning Rate Decay | Shrinks by 0.1 at 150th, 180th, and 210th epochs |
| Momentum | 0.9 |
| Weight Decay | 5e-4 |
| CE Loss Weight | KD, CTKD: 0.1
DKD, MLKD: 1.0 |
| Temperature ($\tau$) | 2 (default) |
| KD Loss Weight ($\lambda_{\text{KD}}$) | 9 (default)
DKD: 12 |

Table 10: Experiment settings on ImageNet

| Parameter | Value |
|---|---|
| Optimizer | SGD |
| Epochs | 100 |
| Batch Size | 512 |
| Learning Rate | 0.2
Divided by 10 at 30th, 60th, and 90th epochs |
| Momentum | 0.9 |
| Weight Decay | 1e-4 |
| CE Loss Weight | KD, CTKD: 0.1
DKD, MLKD: 0.5 |
| Temperature ($\tau$) | 2 (default) |
| KD Loss Weight ($\lambda_{\text{KD}}$) | 9 (default)
DKD: 12 |

methods, we use the traditional KD method (Hinton et al., 2015), with a teacher-student setup of T5-Base/T5-Small with parameters 222M/60M.

**Implementation Details.** We use the AdamW optimizer (Loshchilov & Hutter, 2017) with a training learning rate of 0.0003 and a distillation learning rate of 0.0005, with weight decay set to 0.1. The number of training epochs is 10, the batch size is 32, and the distillation coefficient and temperature are set to 0.3 and 4.0, respectively. The rank and number of expert heads of InherNet are set to 128 and 2, respectively, with a total of 128M parameters.

### C.1.3 VISION ⇔ LANGUAGE: MULTI-MODAL EVALUATION

We use the AdamW optimizer (Loshchilov & Hutter, 2017) with a learning rate of 0.001 and a weight decay of 0.1. A cosine learning rate schedule is applied with a linear warm-up for 10K iterations over 30 epochs. The batch size is set to 1024, and the hyperparameters for CLIP-KD are consistent with those in the original paper (Yang et al., 2024), while other training settings follow the original CLIP (Radford et al., 2021). It is worth noting that, due to limited computational resources, our experiments were pre-trained only on CC3M, with all other experimental settings aligned with those in the original CLIP-KD paper. Following standard practices, we use Recall@K (R@K) to evaluate

Table 11: Configuration of paired visual and text encoders used in multimodal evaluation.

| Visual encoder: CNN | | Text encoder: Transformer | | | |
|---|---|---|---|---|---|
| Model | Params | Layer | Width | Head | Params |
| ResNet-101 | 56.26M | 12 | 512 | 8 | 37.83M |
| ResNet-18 EfficientNet-B0 MobileNetV3 | 11.43M 4.66M 2.04M | 12 | 384 | 6 | 21.29M |
| InherNet | 12.74M | 12 | 512 | 8 | 18.95M |

Table 12: Comparison of top-1 and top-5 Accuracy (%) on the ImageNet validation set between InherNet and previous state-of-the-art knowledge distillation methods, using ResNet34 as the teacher and ResNet18 as the student.

| Accuracy | Teacher | Student | CRD | OFD | ReviewKD | SimKD | CAT-KD | KD | KD+CTKD | DKD | MLKD | MLKD+Logit Std. | InherNet$_{Small}$ | InherNet$_{Large}$ |
|---|---|---|---|---|---|---|---|---|---|---|---|---|---|---|
| top-1 | 73.31 | 69.75 | 71.17 | 70.81 | 71.61 | 71.59 | 71.26 | 71.03 | 71.38 | 71.70 | 71.90 | 72.08 | 71.83 | **72.46** |
| top-5 | 91.42 | 89.07 | 90.13 | 89.98 | 90.51 | 90.48 | 90.45 | 90.05 | 90.27 | 90.41 | 90.55 | 90.74 | 90.73 | **90.88** |

retrieval performance within the top K nearest neighbors. Table 11 shows the configurations of the visual and text encoders used in multimodal evaluation.

## C.2 RESULTS ON IMAGENET

Table 12 presents a comparison of top-1 and top-5 accuracy between InherNet and various distillation methods on the ImageNet dataset. It can be observed that InherNet $_{Large}$ continues to demonstrate leading performance on the large-scale dataset, while InherNet $_{Small}$ achieves a good balance between efficiency and effectiveness.

## C.3 ADDITIONAL EXPERIMENTS ON LARGE TRANSFORMER LANGUAGE MODELS

The T5 experiments in the main text demonstrate that InherNet naturally extends to encoder–decoder Transformers Raffel et al. (2020). To further assess the applicability of InherNet to large-scale autoregressive LMs, we conduct additional experiments on LLaMA-2-7B Touvron et al. (2023), using its HuggingFace implementation.

**Why conventional KD is difficult for LLaMA-style LMs.** Token-level KD for large decoder-only LMs faces several practical challenges:

- **High-dimensional logits.** LLaMA-2 uses a vocabulary of roughly 32K tokens. Token-level KD requires the teacher to output the full vocabulary distribution at every generation step and the student to match it via a KL loss. This leads to large tensors of shape $32k \times$ (sequence length) $\times$ (batch size), which makes KD memory- and bandwidth-intensive, especially for long sequences.

- **Strict teacher forcing.** For stable token-level KD, the teacher must read ground-truth tokens when producing logits. If the teacher instead conditions on student-generated tokens, the KL targets become unstable. However, teacher forcing at each time step requires forwarding the teacher model for every position in the sequence, which increases both memory usage and computation.

- **Tokenizer mismatch.** Standard KL-based KD assumes that the teacher and student share the same vocabulary and output dimension. When the tokenizers differ, the teacher's probability vector over its vocabulary cannot be directly aligned with the student's vocabulary, and the KL loss is no longer well-defined. In practice, KD is thus only straightforward when both models share an identical tokenizer and output layer.

- **Empirical instability.** Recent studies on KD for autoregressive LMs report that traditional KD can degrade performance or fail to converge in some regimes Gu et al. (2023); Zhong et al. (2024), further limiting its applicability to large LLMs.

In contrast, our proposed InherNet does not rely on a token-level KL loss between teacher and student outputs. Instead, it operates directly at the level of network weights: we factorize and inherit the linear layers of the teacher using the proposed SVD-based initialization and asymmetric MoE

Table 13: Additional results on LLaMA-2-7B. "Math" reports accuracy on GSM8K; "Generality" reports accuracy on an MMLU-style evaluation. The InherNet achieves comparable or better performance than the 7B teacher in math reasoning while using substantially fewer parameters.

| Method | Parameters | Math | Generality |
|--------|-----------|------|-----------|
| Teacher | 7B | 76.34 | 50.31 |
| Student | 7B | 76.88 | 51.06 |
| InherNet | 4.2B | 77.19 | 49.82 |

structure. As a result, InherNet is agnostic to the exact vocabulary and can be applied even when the teacher and student have different tokenizers.

We implement InherNet on LLaMA-2-7B, setting the number of experts to $H = 4$ and the rank of each inherited linear mapping to $r = 256$. We consider two domains:

- **Math reasoning.** We fine-tune on MetaMathQA Yu et al. (2023) and evaluate on GSM8K Cobbe et al. (2021).
- **General instruction following.** We fine-tune on databricks-dolly-15k Conover et al. (2023) and evaluate on an MMLU-style benchmark Hendrycks et al. (2020).

Conventional KD baselines are not applicable due to the aforementioned constraints. Instead, we include a self-distillation baseline where both teacher and student share the same LLaMA-2-7B weights. This setting is equivalent to Zhang et al. (2019), and thus offers no parameter compression. For InherNet, we compress the 7B-parameter teacher into a 4.2B-parameter InherNet model. The results are summarized in Table 13.

**Discussion.** The 4.2B-parameter InherNet model slightly surpasses the original 7B teacher on GSM8K and maintains competitive performance on the general instruction-following benchmark. This indicates that InherNet is also applicable to larger-scale Transformer LMs, achieving strong performance with nearly half the parameter count. We attribute this behavior to the extended SVD initialization, which preserves most of the teacher's forward computation, and the asymmetric MoE structure, which supports stable generalization during fine-tuning.

**Summary of scalability to larger models** The above experimental results already evaluate InherNet on substantial-scale models and datasets. To further demonstrate scalability to large decoder-only Transformer LMs, we instantiate InherNet on LLaMA-2-7B, following the above setup (Table 13). In this experiment, the teacher and the self-distilled baseline both use the full 7B-parameter LLaMA-2 model, whereas the inherited model compresses the teacher into a 4.2B-parameter Inher-Net. The inherited model matches or slightly surpasses the teacher.

These results, together with the architecture-agnostic nature of our theory (Theorem 2.4, Theorem B.5, and Corollary 2.9), indicate that InherNet is not limited to small networks: the same SVD-based inheritance and asymmetric MoE structure can be applied to larger convolutional, Transformer, and multimodal architectures, and remains effective in large-model regimes.

**InherNet's performance on more complex multimodal tasks** We have conducted additional experiments on the VQA task using VQA v2.0 Goyal et al. (2017). With the teacher/student models MobileVLM V2 3B[1]/1.7B[2] Chu et al. (2024), we follow Chu et al. (2024) by freezing the vision encoder and tokenizer, and splitting training into a pre-training stage and a multi-task fine-tuning stage. For top-K selection in text-focused vision-token knowledge distillation, we take the top 16 tokens with the highest attention scores. In the multi-task stage, only VQA data is used. All other settings follow Chu et al. (2024). Given the dataset's large scale and limited computational resources, which make training excessively time-consuming, we train the model on only 10% of the training set and evaluate cross-modal alignment on the full test set. We compare the proposed InherNet with vanilla KD Hinton et al. (2015), setting the number of experts $H = 4$ and rank $r = 256$. Results are shown below:

---

[1] https://huggingface.co/mtgv/MobileVLM_V2-3B
[2] https://huggingface.co/mtgv/MobileVLM_V2-1.7B

Table 14: Comparison of VQA performance between vanilla KD and InherNet.

| Method | Size | overall | other | number |
|---|---|---|---|---|
| Vanilla | 1.7B | 56.12 | 38.45 | 34.28 |
| InherNet | 1.5B | 57.03 | 41.63 | 37.22 |

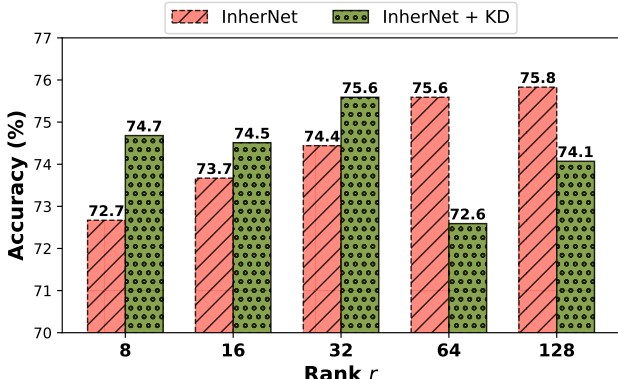

Figure 3: The impact of distillation on InherNet of different scales with varying ranks.

## C.4 EXPERIMENTAL DETAILS OF INSIGHTS ON INHERNET

**Insight 1. Distillation benefits small-scale InherNet, but harms the performance of large-scale InherNet.**

We construct InherNets of different scales based on various ranks ($r = 8, 16, 32, 64, 128$), using them as student networks, and adopt ResNet-56 as the teacher network for knowledge distillation (Hinton et al., 2015). Figure 3 shows the performance changes of InherNets of different scales before and after distillation. The results indicate that small-scale InherNets ($r \leq 32$) benefit significantly from distillation, with smaller models showing greater improvement. In contrast, large-scale InherNets experience a performance drop after distillation. We speculate that this is because large-scale InherNets already inherit most of the teacher's knowledge and may even outperform the teacher through independent training, InherNet $_{Large}$ consistently outperforms the teacher network). Therefore, further distillation offers little benefit to such networks and may even have a negative effect.

**Insight 2. Rank significantly affects model performance; using multiple expert heads performs better than a single head.**

We investigate the impact of rank size $r$ and the number of expert heads $H$ on the performance of InherNet. As shown in Figure 4, rank exerts a more significant influence on InherNet. We attribute this to the fact that rank directly determines how well InherNet inherits core knowledge from the teacher network, which is crucial for subsequent learning—an advantage that cannot be compensated by simply increasing model parameters such as the number of expert heads. Moreover, we also observe that single-head structures perform worse than multi-head ones. However, increasing the number of heads beyond one ($H > 1$) does not necessarily lead to further gains. We speculate that while multi-head mechanisms combined with gating improve generalization and diversity, too many expert heads may cause overfitting.

**Insight 3. Compared to traditional distillation methods, InherNet converges significantly faster.**

We select the commonly used KD method with epoch=240 as the subject for studying model convergence speed. Figure 5 shows the change in training and testing loss during the training process for different methods in a single experiment. The results indicate that InherNet converges significantly faster than traditional KD methods. However, InherNet without SVD initialization shows highly unstable generalization performance during training. This phenomenon strongly supports the effectiveness of the proposed extended SVD method in accelerating model convergence.

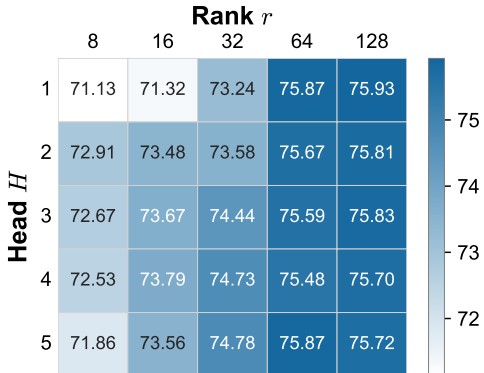

Figure 4: The impact of rank size $r$ and the number of expert heads $H$ on the performance of the proposed InherNet.

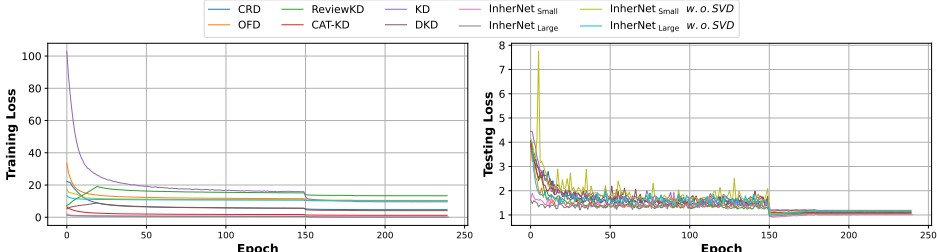

Figure 5: The training and testing loss curves of InherNet and various KD methods during training.

# D    RELATED WORK

## D.1    KNOWLEDGE DISTILLATION

Knowledge distillation (KD) (Hinton et al., 2015) is an efficient model compression technique that helps a smaller student network learn better during training by transferring the knowledge of a larger teacher network. Traditional knowledge distillation methods typically focus on minimizing the prediction differences between the teacher and student networks, typically using loss functions like KL divergence calculated from the teacher's logit outputs. ecent research has focused on two main directions: logit distillation (Hinton et al., 2015; Chen et al., 2020; Li et al., 2020; 2023b; Zhao et al., 2022; Jin et al., 2023; Zhang et al., 2023a; Sun et al., 2024) and feature distillation (Yim et al., 2017; Peng et al., 2019; Heo et al., 2019; Chen et al., 2021; Li et al., 2021; Chen et al., 2022; Lin et al., 2022; Guo et al., 2023). In logit distillation, the goal is to optimize the similarity of the outputs by matching the prediction probabilities of the teacher and the student, while feature distillation focuses on transferring the intermediate feature representation within the model, providing more fine-grained learning information. However, due to the limited capacity of the student network, especially the performance ceiling imposed by the teacher network, the student's performance often struggles to match that of the teacher (Gou et al., 2021; Hu et al., 2023; Xu et al., 2024b).

## D.2    LOW-RANK DECOMPOSITION

In recent years, low-rank decomposition has become a key focus in model compression, aiming to reduce storage requirements and improve inference efficiency by approximating weight matrices through the product of low-rank matrices. Initially applied to the weight matrices in convolutional neural networks (CNNs), particularly the 4D tensors of convolutional kernels, early studies (Denton et al., 2014; Lebedev et al., 2014; Tai et al., 2015; Xu et al., 2019; Yang et al., 2020) focused on effective matrix and tensor decomposition methods, such as using CP decomposition to decompose convolutional kernels into multiple low-rank layers (Lebedev et al., 2014). While this approach reduces model complexity, it can lead to issues like vanishing or exploding gradients when scaling to large, deep networks, making training and fine-tuning more challenging (Tai et al., 2015).

To address this, subsequent research has reshaped high-dimensional tensors into 2D matrices and applied techniques like singular value decomposition (SVD) for compression, effectively reducing computational load. For example, channel decomposition (Zhang et al., 2015) using SVD reduces convolution layers to two, with one layer being a $1 \times 1$ convolution, and further eliminates small singular value channels to reduce computation. However, these low-rank decomposition methods are typically applicable to specific CNN structures and vision tasks (Wang et al., 2026). Moreover, existing methods have not explored the potential link with student networks in KD, while our proposed InherNet aims to establish an effective bridge between the two.

**Low-precision low-rank compression.** Recent work has explored randomized low-precision low-rank factorization for compressing individual matrices and large language models. Saha et al. (2023) propose LPLR, a general matrix-compression algorithm that approximates a single matrix $A$ as $A \approx LR$ using randomized sketching and joint low-rank and low-precision factors, with approximation-error guarantees. Saha et al. (2024) builds on this idea and applies low-rank and low-precision decompositions to all weight matrices in an LLM, yielding a post-training compression scheme that focuses on bit- and rank-efficient storage while keeping the network architecture unchanged.

In contrast, our work is not a generic matrix-compression routine but an SVD-driven network inheritance framework: we reparameterize the teacher model into a low-rank, gated one-down-many-ups module and train the resulting InherNet end-to-end. Our analysis centers on inheritance-aware architecture and optimization, including convergence guarantees, parameter-efficiency bounds, and expert specialization, which is theoretically instructive and complementary to Saha et al. (2023) and Saha et al. (2024).

## E  THE USE OF LARGE LANGUAGE MODELS

A Large Language Model (LLM) provided assistance in the preparation of this manuscript. Its role was strictly limited to:

- Proofreading: Checking the text for spelling errors and grammatical accuracy.
- Improving Clarity: Suggesting alternative phrasing to enhance readability and flow, without changing the scientific meaning.

All changes suggested by the tool were manually reviewed and implemented by the authors to maintain the integrity of the work.

