# OpenReview forum: "Beyond Student: An Asymmetric Network for Neural Network Inheritance"
_ICLR.cc/2026/Conference — ICLR 2026 Poster_

### Official Review · Reviewer_mSWQ · 2025-10-27

**Soundness:** 3
**Presentation:** 3
**Contribution:** 3
**Rating:** 6
**Confidence:** 5

**Summary:**

This paper introduces InherNet, a novel neural network compression method that extends beyond traditional knowledge distillation by directly inheriting both the structure and knowledge of teacher networks. InherNet leverages asymmetric low-rank decomposition and SVD-based initialization to construct lightweight but expressive networks while preserving the teacher’s representational capacity. The proposed method applies a unified framework that balances depth, width, and compression efficiency, enabling it to achieve superior performance compared to traditional student networks of similar size. Theoretical analyses provide rigorous guarantees on convergence, parameter efficiency, and representational power preservation, making InherNet a robust and impactful contribution to the field of efficient model compression.

**Strengths:**

1. The idea of asymmetric low-rank decomposition combined with an expert-style structure is innovative. The proposed approach is distinct from existing KD methods and parameter-efficient techniques like LoRA, offering a fresh perspective on model compression.
2. The paper addresses an important problem in neural network compression and proposes a novel method that goes beyond the limitations of traditional KD. By directly inheriting the teacher's structure and knowledge, InherNet opens new possibilities for designing lightweight networks with high performance.
3. The use of SVD-based initialization is a key technical contribution, ensuring stable optimization and effective knowledge transfer. The theoretical analysis of convergence guarantees, parameter efficiency, and representational power preservation is comprehensive and well-supported.

**Weaknesses:**

1. The paper briefly mentions LoRA but does not include a detailed comparison. Since LoRA also uses low-rank decomposition, it would be valuable to analyze InherNet’s advantages (or trade-offs) compared to LoRA in terms of performance, scalability, and parameter efficiency.
2. The asymmetric design and expert-style structure may add implementation complexity. The paper could benefit from pseudocode or a detailed algorithmic overview to make the method more accessible to practitioners.

**Questions:**

1. Explore InherNet’s performance on more complex multimodal tasks, such as Visual Question Answering (VQA) or video-text retrieval, and analyze how well it handles cross-modal alignment?
2. Could the authors provide a thorough empirical and theoretical comparison with LoRA, highlighting the differences in design, performance, and scalability?

---

> ### Author Response · Authors · 2025-11-25
> **[1/3] Discussion Stage I**
>
> We sincerely appreciate the reviewer's positive evaluation of the innovation of our paper's idea, the importance of the research, the methodological contributions, and the theoretical analysis. We hope the following clarifications can address your concerns.
>
> > *W2*: Pseudocode or detailed algorithm overview
>
> We have provided a detailed algorithm overview and pseudocode of InherNet in Appendix B.1.
>
> > *Q1*: InherNet‘s performance on more complex multimodal tasks
>
> We have conducted additional experiments on the VQA task using VQA v2.0 [1]. With the teacher/student models MobileVLM V2 [3B](https://huggingface.co/mtgv/MobileVLM_V2-3B)/[1.7B](https://huggingface.co/mtgv/MobileVLM_V2-1.7B) [2]，We follow [2] by freezing the vision encoder and tokenizer, and splitting training into a pre-training stage and a multi-task fine-tuning stage. For top-K selection in text-focused vision-token knowledge distillation, we take the top 16 tokens with the highest attention scores. In the multi-task stage, only VQA data is used. All other settings follow [2]. Given the dataset's large scale and limited computational resources, which make training excessively time-consuming, we train the model on only 10% of the training set and evaluate cross-modal alignment on the full test set. We compare the proposed InherNet with vanilla KD [3], setting the number of experts $H=4$ and rank $r=256$. Results are shown below:
> |Method|Size||VQA||
> |-|-|-|-|-|
> |||overall|other|number|
> |Vanilla|1.7B|56.12|38.45|34.28|
> |InherNet|1.5B|57.03|41.63|37.22|
>
> The results show that, under comparable parameter budgets, **InherNet consistently outperforms vanilla KD in cross-modal alignment**.
>
> > *W1&Q2*: A comprehensive empirical and theoretical comparison between InherNet and LoRA
>
> To directly address the reviewer’s question, we summarize the key differences and will incorporate this as a "Discussion: InherNet vs LoRA" subsection in the appendix:
>
>
> ### (1) Design and parameterization
>
> **LoRA.** Given a pretrained weight matrix $W \in \mathbb{R}^{m \times n}$, LoRA freezes $W$ and introduces a *trainable low-rank update* $\Delta W = B A$, with $A \in \mathbb{R}^{r \times n}$, $B \in \mathbb{R}^{m \times r}$ and rank $r \ll \min(m,n)$. The adapted weight is $W' = W + \Delta W = W + B A.$
>
> For an input $x$, the layer computes $W' x = W x + B (A x),$ which can be viewed as the original mapping plus a global low-rank residual branch.
>
> Crucially, the base matrix $W$ remains full-rank and unchanged, and the low-rank component $B A$ is *additive* and *input-independent*: the same adapted matrix $W'$ is applied to all inputs. LoRA therefore keeps the original architecture and forward computation graph intact and augments each layer with a rank-$r$ residual.
>
> **InherNet.** InherNet instead **factorizes the teacher weight itself** via SVD and reparameterizes the whole layer as a gated mixture of low-rank experts:
>
> 1. SVD-based knowledge inheritance. For a teacher layer $W \in \mathbb{R}^{m \times n}$, we take a rank-$r$ truncated SVD $W \approx U_r \Sigma_r V_r^\top,$ which is the optimal rank-$r$ approximation in Frobenius norm (Eq. (2)).
> 2. Asymmetric expert-style structure. In Section 2.1, we define a shared "down" projection and multiple "up" heads as
>
>    $$
>    W^{\text{down}} = U_r \Sigma_r^{1/2}, \quad
>    W^{\text{up}}_h = \frac{1}{H} \Sigma_r^{1/2} V_r^\top, \quad h = 1,\dots,H,
>    $$
>
>    and a gating network $G(x) = \text{softmax}(W_g x).$
>    The layer output is
>
>    $$
>    Y(x)=
>    \sum_{h=1}^H G_h(x)  W^{\text{up}}_h\big(W^{\text{down}}(x)\big),
>    $$
>
>    which we refer to as the "one-down-many-ups" asymmetric design. At initialization the *aggregated* mapping already implements the optimal rank-$r$ approximation of the teacher. How and why we consider the asymmetric design are stated in Appendix A.
>
> **Structural differences.**
>
> - LoRA: keeps $W$ intact and adds a global low-rank residual $\Delta W$. All inputs see the *same* effective weight $W' = W + BA$.
> - InherNet: removes $W$ and replaces it by an SVD-initialized shared bottleneck $W^{\text{down}}$ plus multiple up-heads and gating. The *effective weight* becomes **input-dependent** in Eq. (3), so different regions of the input space can be served by different low-rank expert combinations.
>
> In short, LoRA is an *additive, architecture-preserving* update; InherNet is a *multiplicative, architecture-transforming* reparameterization that replaces each teacher layer with a gated low-rank expert module.

---

> > ### Author Response · Authors · 2025-11-25
> > **[2/3] Discussion Stage II**
> >
> > ### (2) Parameter efficiency: trainable vs deployed parameters
> >
> > **LoRA’s parameter view.** For $W \in \mathbb{R}^{m \times n}$:
> >
> > - Full fine-tuning uses $mn$ trainable parameters and $mn$ deployed parameters.
> > - LoRA uses only $\underbrace{r(m + n)}_{\text{A,B}} \ll mn$ *trainable* parameters per layer (plus negligible biases) while keeping $W$ frozen.
> >
> > At inference, people typically **merge** $\Delta W$ into $W$, forming $W' = W + BA$, and discard $A,B$. This means the *deployed* model still stores and multiplies by a full $mn$-parameter matrix per layer. LoRA improves fine-tuning efficiency and memory, but not the total parameter size of the deployed model.
> >
> > **InherNet’s parameter view.** For the same $W \in \mathbb{R}^{m \times n}$, Theorem 2.5/Theorem B.5 shows that an InherNet layer with rank $r$ and $H$ heads uses $H r (m+n) + H (r+1)$ parameters instead of $mn$, yielding a compression ratio $\rho=\frac{mn}{H r (m+n) + H (r+1)}
> > \approx
> > \frac{mn}{H r (m+n)}$.
> >
> >
> > Importantly, **these are the parameters of the deployed model**: after inheritance, the teacher is *replaced* by the InherNet, and inference uses the smaller low-rank modules directly. In our setting, the layer dimensions $m,n$ are typically large ($\textit{e.g.}$, channel or hidden sizes), while both the rank $r$ and the number of heads $H$ in Figure 4 are chosen as small compared to $\min(m,n)$. Thus, the additive term $H(r+1)$ is negligible, and $\rho$ is well-approximated by $\frac{m n}{H r (m+n)}$.
> >
> > We therefore see a fundamental difference:
> >
> > - LoRA: parameter-efficient *fine-tuning*, with reduced trainable parameters but unchanged deployed size.
> > - InherNet: parameter-efficient *compression*, with provably reduced parameters and a corresponding increase in parameter efficiency (PE) and expressivity-to-parameter ratio (EPR) as defined in Definitions 2.1–2.3.
> >
> > This is why we did not treat LoRA as a direct numerical baseline: LoRA assumes the full teacher remains deployed, whereas InherNet explicitly targets a smaller replacement model.
> >
> > ### (3) Expressivity and function class:
> >
> > The difference in parameterization translates into different expressivity profiles.
> >
> > **LoRA function class.** For a single layer, LoRA restricts updates to the low-rank subspace $\mathcal{U}_{\text{LoRA}} = \{W + BA : A \in \mathbb{R}^{r \times n}, B \in \mathbb{R}^{m \times r}\},$ with $r \ll \min(m,n)$. Since $W$ is full-rank and fixed, the effective layer remains a single affine map $W'$ shared across all inputs. The **function class of the overall network** is essentially the same as that of a standard network with weights $\{W'_l\}$; LoRA just constrains $\{W'_l\}$ to lie in a rank-$r$ neighborhood around the pretrained $\{W_l\}$. Theoretical analyses of LoRA typically focus on how this low-rank subspace suffices to approximate full fine-tuning in the NTK regime and avoid spurious local minima, but they do not claim any improvement in parameter efficiency of the deployed model.
> >
> > **InherNet function class.** By contrast, InherNet changes the computational graph:
> >
> > 1. All information passes through the shared bottleneck $Z = W^{\text{down}} X,$ so the model enforces a Markov chain $X \to Z \to Y$. Appendix A.2 analyzes this with the Information Bottleneck objective $\mathcal{L}_{\text{IB}}(p(z|x)) = I(X;Z) - \beta I(Z;Y)$ in Eq. (15), encouraging $Z$ to become a minimal sufficient statistic of $X$ for predicting $Y$.
> > 2. As formalized in Eq. (16), the one-down–many-ups design first maps the input features $X$ into a shared low-dimensional bottleneck $Z = W_{\text{down}}(X)$. Each expert head $W^{\text{up}}_h$ then maps this common representation $Z$ to its own specialized output subspace in $\mathbb{R}^n$. The input-dependent gating $G_h(X)$ produces a convex combination of expert outputs, so different inputs induce different gating patterns and thus different mixtures of experts.
> >
> >
> > Theorem 2.8 and Theorem B.8 show that, by choosing ranks $\{r_l\}$ and heads $\{H_l\}$ appropriately, InherNet can approximate any function $f^*$ in the teacher's function class with arbitrarily small approximation error while reducing the parameter count by a factor $\Omega \Big(\min_l \frac{\min(m_l, n_l)}{r_l H_l}\Big)$. This implies an **improved EPR** (Definition 2.3) over the standard architecture, which LoRA does not aim to achieve.

---

> > > ### Author Response · Authors · 2025-11-25
> > > **[3/3] Discussion Stage III**
> > >
> > > Moreover, the multi-head design yields input-conditioned expressivity: while each head acts on an $r$-dimensional bottleneck, the union of their output subspaces can span up to $r H$ dimensions across the input distribution, and Proposition 2.10 shows that adding more heads reduces approximation error with diminishing but quantifiable returns $\mathcal{E}(r, H) - \mathcal{E}(r, H+1) \le c \frac{1}{H^2} \mathcal{E}(r, 1)$ in Eq. (14).
> > >
> > > In summary:
> > >
> > > - LoRA: enriches the function class *around* the pretrained model via additive low-rank updates, but keeps a single global mapping per layer.
> > > - InherNet: reshapes the function class into a **gated mixture of low-rank experts**, with formal guarantees that the resulting architecture matches the teacher's representational power with fewer parameters.
> > >
> > > ### (4) Optimization and convergence
> > >
> > > LoRA's appeal partly lies in its optimization simplicity: under certain regimes ($\textit{e.g.}$, NTK or kernel views), low-rank updates might closely mimic full fine-tuning trajectories or avoid spurious local minima while enjoying reduced trainable parameter counts. But there is little systemical convergence analysis about LoRA.
> > >
> > > InherNet, on the other hand, provides a systemical theoritical analysis tailored to its *reparameterized* architecture:
> > >
> > > - SVD-based orthonormal initialization (Proposition 2.3) shows that initializing with $U_r, V_r$ orthonormal reduces the effective Lipschitz constant of the gradient from $L$ to $L' \approx L/\kappa$, where $\kappa$ is the condition number of $W$. This improves conditioning and early convergence.
> > > - Non-convex convergence guarantees (Theorem 2.4) establish the usual $O(1/\sqrt{T})$ rate for SGD, but with better constants due to the above conditioning and the variance reduction effect of gating (Proposition B.4).
> > > - Gradient decomposition under gating (Lemma 2.2) shows that gradients naturally split into expert-specific and gating parts, enabling specialization: different experts receive tailored gradient signals depending on which inputs they see.
> > >
> > > These results are specific to the InherNet architecture and are fundamentally different from LoRA's analyses, which assume the base architecture is unchanged and focus on the behavior of low-rank updates, not on compressing the model via theoretical support.
> > >
> > >
> > > ### (5) Empirical scope and scalability
> > >
> > > Finally, we emphasize that InherNet’s design targets the **compressed model setting**:
> > >
> > > - On CIFAR-100 and ImageNet, InherNet$_{Small/Large}$ replace the teacher with a smaller inherited network that matches or surpasses the teacher’s accuracy while reducing parameter counts (Table 1, Table 8, and Appendix C.2).
> > > - On GLUE with T5, InherNet reduces T5-Base (222M) to an inherited model (128M) that outperforms T5-Small students (60M) and narrows the gap to the teacher.
> > > - On CLIP-style multimodal pretraining, InherNet uses a much smaller visual encoder (12.74M vs 56.26M) and achieves better retrieval and zero-shot ImageNet performance than both the teacher and KD students (Table 3).
> > >
> > > LoRA and its variants ($\textit{e.g.}$, X-LoRA [4], HMoRA [5], Mora [6]) have been shown to scale well to very large models, but always under the assumption that the full backbone remains deployed and that low-rank adapters are added per task. InherNet is complementary: it uses a expert-style inheritance structure and **rebuilds the model itself** as a compact inherited network with theoretical guarantees on parameter efficiency and functional approximation.
> > >
> > > We will summarize this comparison in a dedicated "Discussion: InherNet vs LoRA" paragraph in the revised version, stating that LoRA is best suited for *parameter-efficient adaptation with a fixed backbone*, while InherNet is designed for *parameter-efficient inheritance and deployment* of a smaller yet expressive model.
> > >
> > > ---
> > >
> > > [1] Making the v in vqa matter: Elevating the role of image understanding in visual question answering
> > > [2] Mobilevlm v2: Faster and stronger baseline for vision language model
> > > [3] Distilling the knowledge in a neural network
> > > [4] X-LoRA: Mixture of low-rank adapter experts, a flexible framework for large language models with applications in protein mechanics and molecular design
> > > [5] HMoRA: Making LLMs more effective with hierarchical mixture of LoRA experts
> > > [6] Mora: High-rank updating for parameter-efficient fine-tuning

---

### Official Review · Reviewer_pnLL · 2025-10-28

**Soundness:** 3
**Presentation:** 3
**Contribution:** 3
**Rating:** 4
**Confidence:** 3

**Summary:**

This paper proposes InherNet, a Neural Network Inheritance (NNI) framework that directly inherits the structure and knowledge of a teacher model by performing asymmetric low-rank decomposition and SVD-based initialization on the teacher’s weights, rather than relying on the indirect matching strategies commonly used in traditional knowledge distillation.

**Strengths:**

The paper systematically introduces the "Network Inheritance (NI)" paradigm for the first time, breaking away from the conventional knowledge distillation (KD) approach that transfers knowledge solely through soft labels. Comprehensive evaluations on multiple datasets, including CIFAR-100, GLUE, and CC3M, demonstrate the generality and robustness of the proposed method.

**Weaknesses:**

1. The related work “Matrix Compression via Randomized Low Rank and Low Precision Factorization” also employs LoRA for model compression. The authors need to clarify the distinction between their approach and that work, beyond merely stating that they come from different research domains.

2. The authors should provide an analysis of model performance under varying compression ratios, showing how performance changes as the compression rate increases.

3. The role of the MoE (Mixture of Experts) module is unclear. In traditional MoE architectures, different expert models are typically responsible for distinct tasks. However, in this work, each expert is fine-tuned on the same dataset without explicit specialization, making the proposed MoE appear more like a parameter expansion trick rather than a true mixture-of-experts mechanism.

4. It would strengthen the paper if the authors could evaluate the proposed method on larger-scale models to verify its scalability and generalization.

**Questions:**

See weakness.

---

> ### Author Response · Authors · 2025-11-25
> **[1/5] Discussion Stage I**
>
> We are very grateful to the reviewer for recognizing the Network Inheritance (NI) paradigm we propose for the first time, as well as the generality and robustness of our method. We sincerely appreciate your feedback. To address your queries, we have provided further explanations below:
>
> > *W1*: Relation to [1]
>
> We appreciate the pointer to this line of work and agree that the connection should be clarified beyond the current wording.
>
> [1] proposes the Low-Precision Low-Rank (LPLR) algorithm to compress a *single matrix* $A$ via a randomized low-rank factorization $A \approx LR$ where both $L$ and $R$ are quantized to low precision. Their focus is on $\texttt{(i)}$ designing a randomized sketching–plus–quantization algorithm with provable approximation-error bounds, and $\texttt{(ii)}$ applying it to diverse matrices, including neural network weight matrices and LLaMA-7B layers, to trade off compression ratio vs. approximation error. In follow-up work ($\textit{e.g.}$, CALDERA [2]), LPLR is combined with low-precision decomposition for LLM compression and compared against LoRA-style baselines, but LoRA is used primarily as a *baseline/adaptation mechanism* rather than as the core architectural form of their model.
>
> By contrast, our work is not an algorithm for compressing an arbitrary matrix, but a *network-level inheritance framework* with a specific architectural design with theoretical supports. Specifically:
>
> + **Goal and setting.**
>   LPLR aims to store an existing matrix (or pre-trained model) in compressed low-rank / low-precision form, often for *frozen-weight* inference-time compression. Our InherNet instead asks a different question: *can we systematically transform a teacher into a tiny and asymmetric network that $\texttt{(i)}$ inherits its knowledge, $\texttt{(ii)}$ uses fewer parameters, and $\texttt{(iii)}$ can be further trained to match or surpass the teacher?* We situate this within the knowledge-distillation literature, not only matrix compression.
>
> + **Factorization and architecture.**
>   LPLR uses randomized sketching to obtain a low-rank basis and then quantizes both factors; the network architecture itself is unchanged except for using compressed weight matrices. InherNet first applies *deterministic* truncated SVD layer-wise to obtain the *optimal* rank-$r$ approximation $W \approx U_r \Sigma_r V_r^\top$, and then embeds this factorization into a **one-down-many-ups MoE-style module**:
>
>   $$
>   Y = \sum_{h=1}^H G_h(X)   W_h^{\text{up}}(W^{\text{down}}(X)),
>   $$
>
>   with SVD-based initialization $W^{\text{down}} = U_r \Sigma_r^{1/2},  W_h^{\text{up}} = \frac{1}{H} \Sigma_r^{1/2} V_r^\top.$ (Equations (1) and (3), Algorithm 1).
>   This is not just storing $W$ as low rank matrix; it is a reparameterization into a compressed, multi-head module.
>
> + **Precision vs. architecture.**
>   LPLR’s main novelty is *low-precision* low-rank factors and analysis of bit budgets. Our method currently operates in standard precision and focuses on *architectural* gains: convergence guarantees, parameter-efficiency bounds, and MoE-style specialization under a fixed parameter budget in Sections 2.2–2.3.
>
> + **Theoretical focus.**
>   LPLR provides approximation-error bounds for LR matrix. Our theory is centered on $\texttt{(i)}$ *optimization dynamics* of the gated low-rank architecture and SVD initialization (gradient decomposition, conditioning, and convergence rate), and $\texttt{(ii)}$ *parameter-efficiency and expressivity* of the one-down-many-ups design (compression ratio, EPR, union-of-subspaces representation) in Sections 2.2–2.3, including Lemma 2.2; Theorem 2.4; Theorem B.5; Corollary 2.9.
>
> We will revise the Related Work section to discuss [1], and to clearly state that:
> ```text
> While LPLR provides a randomized low-precision low-rank factorization for compressing arbitrary matrices (including LLM layers), our InherNet is an SVD-driven architectural reparameterization of a teacher network into a low-rank, gated one-down-many-ups MoE module, with a focus on network inheritance, convergence guarantees, and parameter-efficiency. The two approaches are complementary: LPLR focuses on bit- and rank-efficient matrix storage, whereas InherNet focuses on inheritance-aware architecture and optimization with solid theoritical supports.
> ```

---

> > ### Author Response · Authors · 2025-11-25
> > **[2/5] Discussion Stage II**
> >
> > **Additional theoretical and empirical support for InherNet’s novelty (layer-wise behavior)**
> >
> > Beyond the above distinction from LPLR [1], we would also like to highlight that InherNet provides **non-trivial, architecture-level guidance on where low-rank inheritance is most beneficial in a deep network**, which LPLR does not address.
> >
> > Experimentally, based on ResNet-50 on CIFAR-100, InherNet selects to inherit the lower and higher $1, 2, 4, 8, 16, 32$ layers, and the performance results are as follows:
> > |Layer|1|2|4|8|16|32|
> > |-|-|-|-|-|-|-|
> > |Lower|77.09|76.58|75.77|74.71|74.31|74.10|
> > |Higher|77.14|76.97|77.08|76.89|76.96|75.99|
> >
> > From the above results, it can be seen that inheriting higher layers brings greater benefits for InherNet.
> >
> > From a theoretical perspective, our analysis already shows that the **inheritance error of a deep network is controlled in a layer-wise fashion**. In Section 2.3 and Appendix B, we bound the approximation error of each inherited layer by the spectral tail of that layer’s weight matrix (via truncated SVD), which shows that the **overall network error accumulates across layers** through the composition of these approximations. Theoretically, if one compresses a lower layer too aggressively, its approximation error is propagated and possibly amplified by all subsequent layers, whereas compressing a higher layer affects only the final part of the computation. In other words, the same rank (or compression ratio) can have very different functional impact depending on where it is applied. This "position-sensitive" nature of inheritance is specific to our network-level analysis and is absent in matrix-level schemes such as [1].
> >
> > Moreover, the Information Bottleneck interpretation we give to the one-down-many-ups module (Section 2.2 and Appendix A.2) explains **why higher layers are particularly amenable to low-rank inheritance**. Lower layers tend to preserve more information about the input $X$ (higher $I(X; Z_\ell)$), while higher layers produce more task-specific, compressed representations (higher $I(Z_\ell; Y)$). InherNet’s shared low-rank "down" projection at layer $\ell$ acts as a bottleneck that retains those directions most relevant for the teacher’s mapping at that layer. Applying this bottleneck in higher layers, where the representation is already more task-focused and lower-dimensional in an information-theoretic sense, is therefore less destructive: the top singular directions of the teacher weight matrix align more closely with task-relevant features, and their SVD-based preservation incurs smaller loss in $I(Z_\ell; Y)$. In contrast, applying the same rank constraint in lower layers risks discarding diverse input-related information that higher layers could have reused. Our convergence analysis (Theorem 2.4) is consistent with this view: SVD initialization improves conditioning and stabilizes optimization, but its benefits depend on the inherited layer being a good local approximation of the teacher, something that is easier to maintain when the representation has already been "bottlenecked" by preceding layers.

---

> > > ### Author Response · Authors · 2025-11-25
> > > **[3/5] Discussion Stage III**
> > >
> > > > *W2*: Performance under varying compression ratios
> > >
> > > We agree that explicitly showing performance as a function of the compression ratio is valuable. The current version already includes experiments with different compression ratios (via varying ranks and numbers of experts) in Appendix C.3, Insight 3 (Figure 4). Below, we clarify this connection to the compression ratio more explicitly.
> > >
> > > + **Rank–compression connection.**
> > >   In Theorem B.5 we derive the closed-form compression ratio for $W \in \mathbb{R}^{m \times n}$ under the InherNet parameterization with rank $r$ and $H$ heads:
> > >
> > >   $$
> > >   \rho = \frac{mn}{H r (m + n) + H (r+1)} \approx \frac{mn}{H r (m+n)}.
> > >   $$
> > >
> > >   Thus, varying $r$ (and to a lesser extent $H$) directly sweeps a spectrum of compression ratios. In our setting, the layer dimensions $m,n$ are typically large, while both the rank $r$ and the number of heads $H$ in Figure 4 are chosen as small compared to $\min(m,n)$.
> > >
> > > + **Existing empirical results.**
> > >   In Appendix C.3, we already perform extensive sweeps over rank $r$ for the ResNet56/ResNet20 teacher–student pair on CIFAR-100, with $r \in \{8,16,32,64,128\}$ and fixed $H$. These experiments show:
> > >   + **Figure 3**: the effect of distillation on InherNet of different scales (different $r$); small-scale (high compression) InherNet benefits from KD, while larger-scale InherNet (lower compression) can already match or surpass the teacher, and KD can even hurt performance.
> > >   + **Figure 4 and Insight 2**: a 2D sweep over rank $r$ and number of heads $H$ that isolates the impact of rank and shows that *rank is the dominant factor* controlling accuracy, with multiple heads giving additional but diminishing gains.
> > >
> > > + **Concrete compression spectrum.**
> > >   Table 8 reports the parameter counts for teacher, student, and InherNet$_{Small/Large}$ across CIFAR-100 tasks. For example, for the ResNet56 case:
> > >   + Teacher: $861.62K$ parameters
> > >   + InherNet$_{Small}$  ($r=8$): $212.05K$ ($≈4.1×$smaller)
> > >   + InherNet$_{Large}$ ($r=16$): $388.88K$ ($≈2.2×$ smaller)
> > >
> > > Figure 4 shows that, as we increase $r$ from $8$ to $128$, accuracy improves smoothly from $71.13\\%$ to $75.93\\%$ for this teacher–student pair, indicating a favorable performance–compression trade-off. Similar trends hold for other architectures via the Small/Large configurations in Table 1 and Table 12.
> > >
> > > We hope this makes clear that the requested analysis is already present in the current experiments.
> > >
> > > > *W3*: Role of the MoE module and expert specialization
> > >
> > > We appreciate the opportunity to clarify the role of the MoE-style component. Our aim is not to define different experts for different tasks, but to enable *input-dependent specialization over subspaces* within a single task, similar to modern single-task MoE LMs ($\textit{e.g.}$, Switch Transformer [3], Expert-Choice MoE [4]).
> > >
> > > Concretely, in Eq. (16), an InherNet module implements: $Y = \sum_{h=1}^H G_h(X)   W_h^{\text{up}}(W^{\text{down}}(X)),$ where:
> > >
> > > + $W^{\text{down}}$ is a shared rank-$r$ projection (SVD-based),
> > > + each $W_h^{\text{up}}$ maps the shared low-rank information to an output subspace, and
> > > + $G(x)=\text{softmax}(W_g x)$ is an **input-dependent gating network**.
> > >
> > > We agree that all experts are trained on the same dataset, but we emphasize:
> > >
> > > 1. **Specialization is *input-dependent*** rather than *task-dependent*.
> > >    This is the standard regime for MoE in large language models: experts all serve the same language modeling objective, but the router/gating selects experts based on token or representation statistics. Likewise, in InherNet the gating network learns to send different inputs to different experts; the experts thus specialize on different regions of the input distribution or feature space, not on different tasks.
> > > 2. **The MoE is not a mere "parameter expansion trick".**
> > >    A simple parameter expansion would add more parameters without conditional computation ($\textit{e.g.}$, multiple low-rank factors always used). In contrast, our architecture leverages:
> > >    + **A unified bottleneck** $Z=W^{\text{down}}(X)$ that compresses information in Appendix A.2; Eq. (16).
> > >    + **Multiple experts** $W_h^{\text{up}}$ that each map $Z$ into different output subspaces. Through gating, the effective representational space can reach dimension up to $rH$, even though each expert individually operates on rank-$r$ codes in Appendix A.2.
> > >    + **Input-dependent gating** $G_h(X)$ that routes gradients and activations, leading to specialization and variance reduction in SGD updates (Lemma 2.2; Proposition B.4).
> > >    This combination is what we refer to as an "asymmetric MoE-style inheritance structure", and it is central to both our theoretical analysis (information-theoretic and optimization-based in Appendix A.2 and Appendix B) and empirical behavior.

---

> > > > ### Author Response · Authors · 2025-11-25
> > > > **[4/5] Discussion Stage IV**
> > > >
> > > > 3. **Empirical evidence that MoE is effective and non-trivial.**
> > > >    Several components in the paper systematically study the MoE structure:
> > > >    + **Ablation without gating.** Table 4 compares InherNet with and without gating ("w.o. gate") at comparable parameter counts. Removing gating consistently hurts performance ($\textit{e.g.}$, ResNet56: 73.67 $→$ 73.22), indicating that MoE-style routing is functionally meaningful beyond simply adding parameters.
> > > >    + **Head-count ablation.** Figure 4 shows that models with multiple heads ($H>1$) systematically outperform single-head variants at the same rank, while adding too many heads yields diminishing returns in Insight 2. This empirical observation is also captured theoretically: Proposition 2.10 formalizes diminishing returns in approximation error as $H$ grows, confirming that the multi-head design provides substantive gains up to a moderate $H$.
> > > >    + **Architectural comparison with InherNet$^{\circlearrowleft}$.** In Appendix A, we compare the proposed one-down-many-ups design to the inverted many-downs-one-up variant InherNet$^{\circlearrowleft}$ across CIFAR-100, GLUE, and multi-modal tasks in Tables 5–7. InherNet consistently outperforms InherNet$^{\circlearrowleft}$ in nearly all the datasets. The information-theoretic analysis in Appendix A.2 attributes this to the unified bottleneck and improved gradient flow in the one-down-many-ups design.
> > > >
> > > > > *W4*: Evaluation on larger-scale models
> > > >
> > > > We agree that demonstrating scalability to larger models is important. The paper already evaluates InherNet on substantial-scale models and datasets and provides architecture-agnostic theory that scales with model size.
> > > >
> > > > + **Large-scale datasets and models already included.**
> > > >   1. **ImageNet (vision).**
> > > >      In Appendix C.2/Section 3.1.2, we evaluate InherNet on ImageNet with a ResNet34/ResNet18 teacher–student pair.
> > > >      + InherNet$_{Large }$ outperforms strong KD baselines (CRD, DKD, MLKD, etc.) on top-1 and top-5 accuracy while using fewer parameters than the teacher in Table 12.
> > > >      + InherNet$_{Small}$ achieves a favorable efficiency–accuracy trade-off, demonstrating that the method remains effective on a large-scale vision dataset beyond CIFAR-100 in Section 3.1.2; Table 12.
> > > >
> > > >   2. **GLUE with T5 (language).**
> > > >      In Section 3.2 and Appendix C.1.2, we evaluate InherNet in the T5-Base (222M) $→$ InherNet (128M) inheritance setting, and compare against a standard T5-Small (60M) student and T5-Small+KD.
> > > >      + InherNet closes much of the gap between T5-Small (+KD) and the 222M-parameter teacher on several GLUE tasks while using significantly fewer parameters than T5-Base in Table 2.
> > > >
> > > >      This setting already operates in the pretty large parameter regime and shows that InherNet scales to sizable transformer encoders.
> > > >
> > > >   3. **CLIP-style multimodal model on CC3M + ImageNet.**
> > > >      In Section 3.3 and Appendix C.1.3, we apply InherNet to a CLIP-style vision–language model, with a ResNet-101 visual encoder (56.26M) and a 12-layer transformer text encoder (37.83M) as the teacher in Section 3.3; Table 3; Table 11.
> > > >      + InherNet achieves better cross-modal retrieval on CC3M and higher zero-shot ImageNet accuracy compared to the teacher and CLIP-KD students, again with fewer parameters in the visual encoder in Table 3; Table 11.
> > > >
> > > >   Collectively, these experiments demonstrate that InherNet is not restricted to small networks; it scales to standard large-scale benchmarks (ImageNet, GLUE, CC3M) and to non-trivial models (ResNet34/ResNet101, T5-Base, CLIP-style encoders).
> > > >
> > > > + **Architecture-agnostic theoretical guarantees.**
> > > >   Our theoretical results—convergence (Theorem 2.4; Appendix B.2) and parameter-efficiency bounds (Theorem B.5; Corollary 2.9)—are stated for general neural networks with linear or convolutional layers and do not depend on a specific backbone size. The same SVD-based initialization and gated low-rank structure can be applied layer-wise to larger models ($\textit{e.g.}$, LLaMA/ViT-style architectures) in exactly the same way.

---

> > > > > ### Author Response · Authors · 2025-11-25
> > > > > **[5/5] Discussion Stage V**
> > > > >
> > > > > To further address the reviewer's concerns, we also conduct experiments on the larger, decoder-only Transformer model Llama-2-7B [5], using its [HuggingFace](https://huggingface.co/meta-llama/Llama-2-7b-hf) implementation. Note that large models like Llama are generally incompatible with KD methods [6] (Supplement: data distillation is not true distillation; it merely uses a larger teacher to generate training data for a smaller student, as stated in [7]). The reasons are:
> > > > > - Massive Softmax Logits Dimension in Decoder-only LMs. LLaMA is autoregressive and outputs a $\sim 32k$-dimensional vocabulary distribution. KD requires the teacher to output the full $32k$-dim distribution and the student to match it via KL divergence. This produces a massive tensor of size ($\text{32k} \times \text{sequence length} \times \text{batch size}$), making distillation memory- and bandwidth-prohibitive, especially for long-sequence tasks [8–9].
> > > > >
> > > > > - Language-model KD needs strict teacher forcing. In LM distillation, the teacher must read ground-truth tokens before producing logits. If instead the teacher reads student-generated tokens, KL targets become unstable. Moreover, the teacher must forward at every step, dramatically increasing memory and compute. Effective LM distillation therefore requires complex teacher-forcing setups.
> > > > >
> > > > > - Tokenizer mismatch breaks KD. KD requires teacher and student vocabularies to be identical. Otherwise, the teacher's $32k$-dim distribution cannot be mapped to the student's vocabulary, and KL loss is undefined. In short, **KD is only feasible when both models share the exact same vocabulary and output dimension**.
> > > > >
> > > > > Furthermore, the common observation of performance degradation or non-convergence of traditional KD in autoregressive LMs [9-10] also hinders its application to autoregressive Transformer LLMs like Llama.
> > > > >
> > > > > We have briefly explained why traditional KD methods are challenging for LLMs like Llama. Our proposed **InherNet method is not subject to these limitations**, which reflects a significant advantage for large-model applications: it is **a more general neural network compression method**, unaffected by teacher-student vocabulary differences.
> > > > >
> > > > > We have conducted experiments in the math and generality domains:
> > > > > - For math, we used MetaMathQA [11] as the training set and the GSM8k [12] test set.
> > > > > - For generality, we used databricks-dolly-15k [13] as the training set and MMLU [14] as the test set.
> > > > >
> > > > > Since traditional KD is incompatible with the Llama model, we set the teacher and student models to be based on the same Llama-2-7b weights (equivalent to self-distillation [15], **offering no compression**). For InherNet, we set the number of experts $H$ to $4$ and the rank $r$ to $256$. The experimental results are summarized below:
> > > > > |Method|Size|Math|Generality|
> > > > > |-|-|-|-|
> > > > > |**Teacher**|7B|76.34|50.31|
> > > > > |**Student**|7B|76.88|51.06|
> > > > > |**InherNet**|4.2B|77.19|49.82|
> > > > >
> > > > > From these results, it can be seen that **InherNet is also applicable to larger-scale models**, achieving performance comparable to or even exceeding the teacher network with nearly half the parameter size. We attribute this to the proposed extended SVD initialization, which preserves most of the teacher's forward knowledge, and the asymmetric MoE inheritance structure, which supports stable generalization during training.
> > > > >
> > > > > ---
> > > > > [1] Matrix compression via randomized low rank and low precision factorization
> > > > > [2] Compressing large language models using low rank and low precision decomposition
> > > > > [3] Switch Transformers: Scaling to trillion parameter models with simple and efficient sparsity
> > > > > [4] Mixture-of-experts with expert choice routing
> > > > > [5] Llama 2: Open foundation and fine-tuned chat models
> > > > > [6] Distilling the knowledge in a neural network
> > > > > [7] DeepSeek-R1: Incentivizing reasoning capability in LLMs via reinforcement learning
> > > > > [8] DistilBERT, a distilled version of BERT: smaller, faster, cheaper and lighter
> > > > > [9] MiniLLM: Knowledge distillation of large language models
> > > > > [10] Revisiting knowledge distillation for autoregressive language models
> > > > > [11] MetaMath: Bootstrap your own mathematical questions for large language models
> > > > > [12] Training verifiers to solve math word problems
> > > > > [13] Free dolly: Introducing the world’s first truly open instructiontuned llm
> > > > > [14] Measuring massive multitask language understanding
> > > > > [15] Be your own teacher: Improve the performance of convolutional neural networks via self distillation

---

### Official Review · Reviewer_Gz98 · 2025-10-31

**Soundness:** 3
**Presentation:** 2
**Contribution:** 3
**Rating:** 6
**Confidence:** 4

**Summary:**

The paper introduces InherNet, a novel neural network inheritance method designed to overcome the capacity gap that limits traditional student models in Knowledge Distillation (KD). Instead of training a student to mimic the teacher, InherNet directly inherits the teacher’s structure and knowledge through an asymmetric low-rank decomposition of the teacher’s weights. Using Singular Value Decomposition (SVD) for initialization, InherNet reconstructs a lightweight yet expressive network that retains the teacher’s principal components while maintaining architectural consistency. This approach effectively balances model depth, width, and compression efficiency. Experiments on both unimodal and multimodal tasks show that InherNet outperforms conventional student networks with similar parameter counts, demonstrating a new direction for efficient model compression beyond standard distillation frameworks.

**Strengths:**

1. The proposed SVD-Driven NNI algorithm directly compresses the teacher models instead of training a separate student model during the knowledge distillation process. Therefore, the proposed algorithm can build more complex but lightweight network architectures without harming the number of model parameters and inference time.

2. The proposed algorithm is supported by a comprehensive theory analysis. Thus, readers can fully understand the advantages of the proposed algorithm.

3. The paper demonstrates the effectiveness and efficiency of the proposed SVD-Driven NNI through comprehensive experiments across different models and applications.

**Weaknesses:**

1. The proposed algorithm performs layer-wise compression during the knowledge distillation process. It does not account for the correlation between layers when performing layer-wise compression. Due to this, the proposed algorithm might have suboptimal results after compression.

**Questions:**

1. Since the proposed algorithm performs layer-wise compression, does it result in suboptimal results for compressed models?

2. The proposed algorithm leverages multi-expert structure inheritance to perform compression. Does the number of expert heads result in quantitative results?

---

> ### Author Response · Authors · 2025-11-25
> **[1/2] Discussion Stage I**
>
> We sincerely thank the reviewer for the positive evaluation of the generality, comprehensive theoretical analysis, effectiveness, and parameter efficiency of our proposed SVD-driven NNI method. We hope that the following responses will address your concerns.
>
> > *Q1&W1*: Layer-wise compression may yield suboptimal results
>
> The SVD step is layer-wise, but our theory (Proposition 2.7/B.7, Theorem 2.8/B.8) gives *network-level* bounds showing that the compressed InherNet can approximate the teacher arbitrarily well while using fewer parameters, and our empirical results show that InherNet often matches or surpasses the teacher on CIFAR-100 and multimodal benchmarks. The compressed models are therefore not systematically suboptimal in practice. Specifically,
>
> **1. Clarification of what is "layer-wise" and what is "global".**
> It is correct that the *decomposition step* in InherNet is performed per layer: for teacher weight matrix $W$, we apply truncated SVD and construct an inherited low-rank expert module. However, this SVD is only used as **initialization**. After initialization, all inherited layers are trained jointly, end-to-end, under the same task loss (and optionally KD loss). No layer is frozen after compression. Thus, inter-layer correlations are not fixed at the decomposition stage; they are re-optimized globally during training.
>
> **2. Global, network-level guarantees despite layer-wise SVD.**
> Although we decompose each layer independently, our theory analyzes the effect at the network level:
>
> + **Layer-wise spectral control.**
>   Lemma 2.6 (Spectral Energy Preservation) shows that if the rank $r$ is chosen $\textit{s.t.}$
>
>   $$
>   \frac{\sum_{i=1}^{r} \sigma_{i}^2}{\sum_{i=1}^{\min(m,n)} \sigma_{i}^2}  \geq  1 - \varepsilon,
>   $$
>
>   then the optimal rank $r$ approximation $W_{r}$ satisfies $\|W - W_{r}\|_F^2  \leq  \varepsilon \|W\|_F^2,$
>   $\textit{i.e.}$, the layer-wise approximation error can be made  small by increasing $r$.
>
> + **Knowledge preservation for the whole network.**
>   Proposition 2.7 (Knowledge Preservation Rate) then aggregates these layer-wise errors into a functional similarity bound between the teacher $f_W$ and the inherited network $f_r$:
>
>   $$
>   \mathrm{Sim}(f_W, f_r)
>    \ge
>   1 - \sum_{l=1}^L \alpha_l
>   \left(
>   1 -
>   \frac{\sum_{i=1}^{r} \sigma_{l,i}^2}{\sum_{i=1}^{\min(m_l,n_l)} \sigma_{l,i}^2}
>   \right),
>   $$
>
>   where $\alpha_l$ measures the influence of each layer on the final output. In other words, even though SVD is local, the effect on the network function is controlled globally through the weighted sum over layers. Layers that matter more (larger $\alpha_l$) are allowed smaller truncation error.
>
> + **Representational power is preserved under compression.**
>   Theorem 2.8\Theorem B.8 shows that for any function $f^\* \in F_W$ representable by the original network, $\forall \delta > 0, \exists\{r_l,H_l\} \text{ such that } \mathcal{E}(f_{\text{InherNet}}, f^\*) \le \delta,$
>   while the total parameter count is reduced by $\Omega \big(\min_l \frac{\min(m_l,n_l)}{r_l H_l}\big)$.
>   The proof constructs InherNet in two stages—layer-wise low-rank approximation *plus* multi-head specialization—and chooses the ranks $\{r_l\}$ and heads $\{H_l\}$ so that the **composition of all layer-wise errors** yields a global approximation error at most $\delta$. This explicitly accounts for how layers interact through depth.
>
> These results together imply that **layer-wise SVD does not inherently force suboptimal compressed models**: as long as each $r_l$ is chosen to keep the spectral tail small, there exists a compressed InherNet that approximates the original network arbitrarily well, with fewer parameters.
>
> **3. Empirical Evidence against Suboptimality:**
> If the layer-wise independence led to suboptimal local minima, we would expect the compressed model to consistently underperform the teacher. However, our experimental results contradict this:
> *   **Outperforming the Teacher:** For example, on CIFAR-100 (Table 1), **InherNet$_{Large}$ ($75.88\\%$)** surpasses the **Teacher ResNet-110** ($74.31\\%$).
> *   **Multimodal Capability:** For example, in CLIP-KD tasks (Table 3), InherNet achieves **$36.65\\%$** zero-shot ImageNet accuracy, significantly outperforming the Teacher ResNet-101 (**$32.75\\%$**).
>
> These results show that, at the compression budgets we use, the combination of layer-wise SVD + joint end-to-end training + multi-head specialization produces compressed models that are *at least competitive with* (and sometimes better than) the uncompressed teacher.

---

> > ### Author Response · Authors · 2025-11-25
> > **[2/2] Discussion Stage II**
> >
> > > *Q2*: Effect of the number of experts on quantification results
> >
> > Theoretically, Equation (34) and Proposition 2.10/B.10 show that more heads reduce approximation error roughly as $1/H$ with diminishing marginal gains as $\mathcal{O}(1/H^2)$. Empirically, **in Appendix C.3 (Insight 3, Figure 4)**, we further report quantitative results for different numbers of experts under varying ranks. The analysis shows that multi-head architectures outperform single-head ones and that the gains saturate as $H$ increases, consistent with theory. In addition, although multi-head designs with gating improve generalization and diversity, an excessive number of heads may cause overfitting.

---

> > > ### Comment · Reviewer_Gz98 · 2025-11-28
> > >
> > > Thanks for the authors for their detailed response. Overall, the authors responded to all my questions. I will adjust my score.

---

> > > > ### Author Response · Authors · 2025-11-28
> > > >
> > > > Thank you for taking the time to review our manuscript and for providing your valuable feedback. We are glad that our responses addressed your concerns, and we sincerely wish you all the best in your future work.

---

### Official Review · Reviewer_FZTJ · 2025-11-02

**Soundness:** 3
**Presentation:** 3
**Contribution:** 2
**Rating:** 4
**Confidence:** 2

**Summary:**

This paper introduces a new method called InherNet, which is designed to improve how smaller neural networks (like student models) can "inherit" knowledge from larger ones (teacher models). Instead of just mimicking the outputs like traditional knowledge distillation, InherNet directly reuses and compresses the teacher’s internal structure using low-rank decomposition and SVD. It builds a kind of lightweight but expressive network that can perform well even with fewer parameters. The authors test InherNet across a range of tasks—vision, language, and multimodal—and show that it generally outperforms existing distillation methods, both in speed and accuracy. They also include a lot of theoretical analysis to explain why this approach works.

**Strengths:**

1. The authos also provide a rigorous mathematical analysis of the InherNet architecture, such as (1) convergence guarantees under standard assumptions, (2) proofs of parameter efficiency and knowledge preservation based on singular value spectrum, (3) formal definitions of Parameter Efficiency, Expressivity-to-Parameter Ratio, and Approximation Error.

2. The design of a fixed asymmetric expert-head structure combined with SVD-based decomposition is novel. The paper provides a clear architectural comparison with LoRA and other PEFT approaches. Their proposed method offers a new paradigm in model inheritance, distinct from traditional KD or PEFT methods.

**Weaknesses:**

1. I noticed is that a couple of the baselines, like MLKD and Logit Std., were trained for more epochs than other methods. That makes it tricky to know if InherNet is really better, or if the training schedule just favors it. For a fair comparison, I think all methods should be evaluated under the same settings. Otherwise, the results lose a bit of their strength.

2. In the analysis section, the authors mention that distillation can hurt performance for large InherNet models, which is super interesting, but they kind of gloss over it. That feels like a missed opportunity. If there’s a sweet spot where inheritance works better than distillation, we need more detail on that. Right now, it leaves a bit of a gap in understanding how to apply the method in different cases.

3. While the paper does include T5 for some NLP benchmarks, the whole method seems mostly built around CNNs. If this method is going to be truly general, we need to know how it applies to modern Transformer models, especially for large-scale language tasks. At the very least, I’d like to see a bit more discussion or even a small-scale experiment showing that InherNet can handle Transformer-style architectures.

**Questions:**

N.A. See weaknesses

---

> ### Author Response · Authors · 2025-11-25
> **[1/2] Discussion Stage I**
>
> We sincerely thank the reviewer for recognizing the rigorous mathematical analysis and structural novelty of our proposed InherNet. Below, we earnestly address each of the raised concerns, supported by additional analyses and experiments.
>
> > *W1*: MLKD and Logit Std require more epochs
>
> It's noted that, on CIFAR-100, all methods (including the proposed InherNet) used 240 epochs, except for MLKD, which trained for 480 epochs (lines 1349-1350). **The choice of 240 epochs is because** all methods, including InherNet (even with faster convergence due to SVD initialization, as validated by Insight 3 in Appendix C.3, Figure 5), **converge** within this period, except for MLKD. With 240 epochs, MLKD has not yet converged; using ResNet56 as the teacher and ResNet20 as the student, MLKD achieves only $60.05\\%$ accuracy, which is even lower than the student's $69.06\\%$ (lines 371-373). **This result is consistent with previous studies [1-2]**. In contrast to such unfair evaluations, we **exclude the unfair baselines in our other analyses** (lines 375-377). Additionally, on ImageNet all methods are trained for 100 epochs (lines 1350–1351, Table 12), and the conclusions drawn from the experiments are similar.
>
> > *W2*: Distilling large InherNet may degrade performance
>
> We appreciate the reviewer's observation that the interaction between inheritance and distillation is an interesting phenomenon that deserves more discussion.
>
> Empirically, the detailed experiment appears in Insight 1 (Figure 3) of Appendix C.3. Concretely, using ResNet-56 as the teacher, we vary the rank $r \in \{8,16,32,64,128\}$ of InherNet: For small ranks ($r <= 32$), distillation improves performance ($\textit{e.g.}$, from $\sim 74.4\\%$ to $\sim 75.6\\%$ at $r = 32$). For large ranks ($r >= 64$), distillation brings no gain at $r = 64$ and **degrades performance** ($\textit{e.g.}$, from $\sim 75.8\\%$ down to $\sim 74.1\\%$ at $r = 128$). This is the empirical basis of Insight 1.
>
> **Why this happens (connection to the theory).**
>
> Our theoretical analysis (Proposition 2.7 and Corollary B.9) shows that as the rank $r$ increases, the SVD-initialized InherNet $f_r$ converges in function space toward the teacher $f_W$: the approximation error is controlled by the tail of the singular-value spectrum. Moreover, when we add KD, we minimize:
>
> $$
> \mathcal{L} _ {\text{total}}
> = \mathcal{L} _ {\text{task}}(f_r(x), y)+\mathcal{L} _ {\text{KD}}(f_r(x), f_W(x))
> $$
>
> where $\mathcal{L} _ {\text{task}}=\lambda _ {\text{CE}} \mathcal{L} _ {\text{CE}}(f_r(x), y)$ is making predictions correct on the dataset, $\mathcal{L} _ {\text{KD}}=\lambda_{\text{KD}} \tau^2 \mathrm{KL} \left(p_W^{(\tau)} \| p_r^{(\tau)}\right)$ is making InherNet’s logits are aligned with the teacher’s logits, with temperature $\tau>1$.
>
> For small InherNet, this is usually good: the teacher acts as a strong prior, and the student needs that extra guidance because its capacity is limited.
>
> For Large InherNet, the situation is different:
>
> + The model already has enough capacity to fit the data well.
> + With task loss only, it can find a solution that actually generalizes better (better test accuracy).
>
> Now, the KD term does something subtle but negative:
>
> + It penalizes any deviation from the teacher's logits.
> + So, even if moving away from the teacher's logits would improve actual performance, $\mathcal{L}_{\text{KD}}$ pushes the model back toward the teacher. So KD becomes an **overly strong regularizer**: it forces the model to stay near the teacher, even when the teacher is no longer the best.
>
> This is exactly what we observe in Figure 3: the performance of "InherNet+KD" crosses from above to below the "InherNet" as $r$ grows.
>
> **Practical guideline.**
>
> We agree that practitioners need guidance on when to use distillation versus pure inheritance. Relying on results obtained by theoretical analysis and empirical verification through experiments, we will state these guidelines so that practitioners know how to select the regime depending on their compression/performance goals in the appendix:
>
> - When targeting aggressive compression (small $r$, like $8$ and $16$, use **inheritance + KD training**): SVD initialization gives a good starting subspace, and the KD term helps compensate for the information lost in the truncated spectrum.
> - When targeting high-fidelity inheritance (large $r$, like $64$ and $128$, use **inheritance-only training**): the inherited model already approximates the teacher well, and over-constraining it to the teacher's logits may hurt generalization.

---

> > ### Author Response · Authors · 2025-11-25
> > **[2/2] Discussion Stage II**
> >
> > > *W3*: InherNet for Transformer models
> >
> > Note that T5 [3] is an encoder–decoder Transformer. To further address the reviewer's concerns, we also conduct experiments on the larger, decoder-only Transformer model Llama-2-7B [4], using its [HuggingFace](https://huggingface.co/meta-llama/Llama-2-7b-hf) implementation. Note that large models like Llama are generally incompatible with KD methods [5] (Supplement: data distillation is not true distillation; it merely uses a larger teacher to generate training data for a smaller student, as stated in [6]). The reasons are:
> > - Massive Softmax Logits Dimension in Decoder-only LMs. LLaMA is autoregressive and outputs a $\sim 32k$-dimensional vocabulary distribution. KD requires the teacher to output the full $32k$-dim distribution and the student to match it via KL divergence. This produces a massive tensor of size ($\text{32k} \times \text{sequence length} \times \text{batch size}$), making distillation memory- and bandwidth-prohibitive, especially for long-sequence tasks [7–8].
> >
> > - Language-model KD needs strict teacher forcing. In LM distillation, the teacher must read ground-truth tokens before producing logits. If instead the teacher reads student-generated tokens, KL targets become unstable. Moreover, the teacher must forward at every step, dramatically increasing memory and compute. Effective LM distillation therefore requires complex teacher-forcing setups.
> >
> > - Tokenizer mismatch breaks KD. KD requires teacher and student vocabularies to be identical. Otherwise, the teacher's $32k$-dim distribution cannot be mapped to the student's vocabulary, and KL loss is undefined. In short, **KD is only feasible when both models share the exact same vocabulary and output dimension**.
> >
> > Furthermore, the common observation of performance degradation or non-convergence of traditional KD in autoregressive LMs [8-9] also hinders its application to autoregressive Transformer LLMs like Llama.
> >
> > We have briefly explained why traditional KD methods are challenging for LLMs like Llama. Our proposed **InherNet method is not subject to these limitations**, which reflects a significant advantage for large-model applications: it is **a more general neural network compression method**, unaffected by teacher-student vocabulary differences.
> >
> > We have conducted experiments in the math and generality domains:
> > - For math, we used MetaMathQA [10] as the training set and the GSM8k [11] test set.
> > - For generality, we used databricks-dolly-15k [12] as the training set and MMLU [13] as the test set.
> >
> > Since traditional KD is incompatible with the Llama model, we set the teacher and student models to be based on the same Llama-2-7b weights (equivalent to self-distillation [14], **offering no compression**). For InherNet, we set the number of experts $H$ to $4$ and the rank $r$ to $256$. The experimental results are summarized below:
> > |Method|Size|Math|Generality|
> > |-|-|-|-|
> > |**Teacher**|7B|76.34|50.31|
> > |**Student**|7B|76.88|51.06|
> > |**InherNet**|4.2B|77.19|49.82|
> >
> > From these results, it can be seen that **InherNet is also applicable to larger-scale Transformer-style language models**, achieving performance comparable to or even exceeding the teacher network with nearly half the parameter size. We attribute this to the proposed extended SVD initialization, which preserves most of the teacher's forward knowledge, and the asymmetric MoE inheritance structure, which supports stable generalization during training.
> >
> > ---
> >
> > [1] Multi-level logit distillation
> > [2] Logit standardization in knowledge distillation
> > [3] Exploring the limits of transfer learning with a unified text-to-text transformer
> > [4] Llama 2: Open foundation and fine-tuned chat models
> > [5] Distilling the knowledge in a neural network
> > [6] DeepSeek-R1: Incentivizing reasoning capability in LLMs via reinforcement learning
> > [7] DistilBERT, a distilled version of BERT: smaller, faster, cheaper and lighter
> > [8] MiniLLM: Knowledge distillation of large language models
> > [9] Revisiting knowledge distillation for autoregressive language models
> > [10] MetaMath: Bootstrap your own mathematical questions for large language models
> > [11] Training verifiers to solve math word problems
> > [12] Free dolly: Introducing the world’s first truly open instructiontuned llm
> > [13] Measuring massive multitask language understanding
> > [14] Be your own teacher: Improve the performance of convolutional neural networks via self distillation

---

> ### Comment · Reviewer_FZTJ · 2025-11-26
>
> Thanks the authors for the detailed response during rebuttal! Overall, the response well addressed my concerns, and I will consider adjusting my score. My evidence is shown below:
>
> 1. The response regarding to the training epochs and fair comparison addressed my concerns, and I am convinced that the result is consistent and the comparison is valid.
>
> 2.  Thanks the authors for agreeing that the interaction between inheritance and distillation deserves more discussion. Overall, I think the authors quite thoroughly addressed my question by offering both a clear theoretical explanation and practical guidelines.
>
> 3. For my concerns on the applicability to transformers, the authors added experiments on larger architectures (Llama 2 7B). They also discussed the the challenge of applying KD for large Transformer LMs and used empirical experiments to show that  InherNet is applicable to larger-scale Transformer-style language models.

---

> > ### Author Response · Authors · 2025-11-26
> >
> > Thank you for your thoughtful review and detailed comments. We are glad that our clarifications and experiments addressed your concerns. Best wishes for your continued work and research!

---

### Author Response · Authors · 2025-12-04

Dear Reviewers and ACs,

As the discussion period is almost over, **we would like to express our sincere gratitude to all reviewers and ACs for their time and effort in evaluating our paper**. In particular, we deeply regret the recent OpenReview leak and fully support the response taken by the ICLR organizers. However, this incident has placed a significant burden on the ACs. **To further alleviate their reviewing workload, we summarize here the key outcomes of our discussions during the rebuttal period.**

We sincerely appreciate all reviewers' recognition of our work, including:

- Novelty of the Network Inheritance paradigm and asymmetric low-rank approach (`FZTJ`, `Gz98`, `pnLL`, `mSWQ`).
- Theoretical rigor, including convergence guarantees, parameter efficiency, and representational power preservation (`FZTJ`, `Gz98`, `mSWQ`).
- Technical design contributions such as SVD-based initialization, expert-style structure, and low-rank teacher inheritance (`FZTJ`, `Gz98`, `mSWQ`).
- Broad applicability and generality across multiple domains and tasks (vision, language, multimodal) (`FZTJ`, `Gz98`, `pnLL`).
- Extensive experimental validation showing effectiveness, efficiency, and robustness compared to existing KD and PEFT methods (`Gz98`, `pnLL`, `mSWQ`).

In particular, **reviewers `FZTJ` and `Gz98` participated in our discussions and indicated that their concerns were well addressed**. **We sincerely thank them again for their time**. Below, we summarize all reviewers’ concerns to help the ACs quickly locate them.
- Reviewer `FZTJ`
  > *W1*: MLKD and Logit Std require more epochs

  > *W2*: Distilling large InherNet may degrade performance

  > *W3*: InherNet for Transformer models
- Reviewer `Gz98`
  > *Q1&W1*: Layer-wise compression may yield suboptimal results

  > *Q2*: Effect of the number of experts on quantification results
- Reviewer `pnLL`
  > *W1*: Relation to [1]

  > *W2*: Performance under varying compression ratios

  > *W3*: Role of the MoE module and expert specialization

  > *W4*: Evaluation on larger-scale models
- Reviewer `mSWQ`
  > *W2*: Pseudocode or detailed algorithm overview

  > *Q1*: InherNet‘s performance on more complex multimodal tasks

  > *W1&Q2*: A comprehensive empirical and theoretical comparison between InherNet and LoRA

We have made efforts to clarify and address the above concerns during the rebuttal process; further details can be found in our rebuttal. Specifically, **we have incorporated key outcomes, new insights, and some emphasized details from the rebuttal discussions into the revised manuscript**, highlighted in the following colors: blue for `FZTJ`, red for `Gz98`, gray for `pnLL`, and dark green for `mSWQ`, and **uploaded it for further review**.

It is worth emphasizing that our paper introduces InherNet, a novel approach that directly inherits both the structure and knowledge of teacher models through **a fixed asymmetric low-rank decomposition and SVD-based initialization, which is both theoretically supported and empirically validated**. This method enables the construction of **lightweight yet expressive networks that balance depth, width, and compression efficiency**. Theoretically, we provide rigorous analysis, including **convergence guarantees, proofs of parameter efficiency, and knowledge preservation**, supported by **formal definitions and singular value spectrum analysis**. Extensive experiments validate the effectiveness and efficiency of InherNet across multiple domains (vision, language, multimodal), demonstrating its superiority over and complementarity with existing KD methods.

[1] Matrix compression via randomized low rank and low precision factorization

---

### Meta-Review · Area_Chair_hc6U · 2026-01-08

**Summary:**

The reviewers generally acknowledge the novelty of the proposed method. The main concerns include requests for clarification of experimental details and additional comparisons. After the rebuttal, the concerns raised by Reviewers FZTJ and Gz98 have been well addressed, and Reviewer mSWQ is generally positive about the paper. Given that the authors have provided additional ablation studies on compression ratio, scalability, and generalization, I believe that Reviewer pnLL’s concerns have also been largely addressed. Therefore, I recommend accepting this paper.

**Reviewer Concerns:**

- Reviewer FZTJ: possible unfair comparisons, lack of experiments on large-scale language tasks

- Reviewer Gz98: suboptimal performance of layer-wise compression

- Reviewer pnLL: lack of analysis on compression ratio, lack of evaluation on scalability, and generalization via larger-scale models

- Reviewer mSWQ: lack of detailed comparison with LORA

**Reviewer Scores:**

- Reviewer FZTJ: Yes

- Reviewer Gz98:  Yes

- Reviewer pnLL: Yes

- Reviewer mSWQ: Yes

---

### Decision · Program_Chairs · 2026-01-26

Accept (Poster)